# The comfort in touch: Immediate and lasting effects of handholding on emotional pain

Razia S. Sahi [1]*, Macrina C. Dieffenbach[1], Siyan Gan[2], Maya Lee[1], Laura I. Hazlett[1], Shannon M. Burns[3], Matthew D. Lieberman[1], Simone G. Shamay-Tsoory[4], Naomi I. Eisenberger[1]

1 Department of Psychology, University of California Los Angeles, Los Angeles, CA, United States of America, 2 School of Education and Psychology, Pepperdine University, Malibu, CA, United States of America, 3 Annenberg School of Communication, University of Pennsylvania, Philadelphia, PA, United States of America, 4 Department of Psychology, University of Haifa, Haifa, Israel

* rsahi1@ucla.edu

**Data Availability Statement:** Data cannot be shared publicly because our IRB states that data may be shared only with principle investigators with verifiable credentials at an academic research

## Abstract

Consoling touch is a powerful form of social support that has been repeatedly demonstrated to reduce the experience of physical pain. However, it remains unknown whether touch reduces emotional pain in the same way that it reduces physical pain. The present research sought to understand how handholding with a romantic partner shapes experiences of emotional pain and comfort during emotional recollection, as well as how it shapes lasting emotional pain associated with emotional experiences. Participants recalled emotionally painful memories or neutral memories with their partners, while holding their partner's hand or holding a squeeze-ball. They additionally completed a follow-up survey to report how much emotional pain they associated with the emotional experiences after recalling them in the lab with their partners. Although consoling touch did not reduce emotional pain during the task, consoling touch increased feelings of comfort. Moreover, participants later recalled emotional memories that were paired with touch as being less emotionally painful than those that were not paired with touch. These findings suggest that touch does not decrease the immediate experience of emotional pain and may instead support adaptive processing of emotional experiences over time.

## Introduction

Three out of four people report that their *most painful* life experience was emotional in nature, rather than physically painful [1]. Emotional pain, defined as an unpleasant feeling (or suffering) associated with a psychological, non-physical origin, often stemming from thwarted psychological needs [2], undergirds a range of psychiatric issues, including depression, anxiety, borderline personality disorder, and suicidal ideation [3, 4]. Given the prevalence of emotional pain, and the negative outcomes associated with such pain, it is crucial to examine how individuals effectively cope with and process it.

When we experience negative events or hardships, support from others can mitigate the harmful effects of those experiences and buffer us from trauma or prolonged distress [5]. For example, talking through our problems or finding a welcome distraction during a tough time

institution. Data access queries can also be sent to the IRB at webIRBHelp@research.ucla.edu. Our data and analysis materials are also hosted on the Open Science Framework for researchers who meet the criteria for access to confidential data: https://osf.io/9wbua/?view_only=3ba7e31d04c54bb9add28a5e5f184c0a.

**Funding:** This work was supported by grant funding from the United States – Israel Binational Science Foundation awarded to Drs. Naomi Eisenberger (N.I.E.) and Simone Shamay-Tsoory (S.S.) (grant number 2015068).

**Competing interests:** The authors have declared that no competing interests exist.

can be valuable in helping us to regulate our emotions [6]. But there are also powerful forms of social support that more implicitly communicate understanding and concern, such as when a loved one holds our hand [7, 8]. This type of physical support, often referred to as consoling touch, is observed across species and across cultures [9], and has been shown to reliably reduce the experience of physical pain [10–12]. Notably, however, research has yet to experimentally assess whether touch *reduces* the subjective experience of emotional pain in the same way that it reduces the subjective experience of physical pain.

Extensive research has documented the importance of physical touch in emotional wellbeing. From a developmental perspective, touch plays a vital role in infants' physical development, neurodevelopment, stress relief, and the development of attachment [13–15]. From a therapeutic perspective, touch is thought to provide comfort and facilitate healing [16–19]. While research suggests that touch can increase positive feelings like security, and decrease negative feelings like stress [19], the majority of research on the pain-relieving effects of touch have focused on how consoling touch affects individuals experiencing physical pain, such as treatment-related pain, or painful shocks administered in experimental settings. This work suggests that handholding, especially with a romantic partner, attenuates subjective distress associated with physical pain, as well as activation in neural regions associated with threat responses [10–12, 20, 21].

A body of research suggests that physical pain and emotional pain are processed in overlapping neural regions [22–24], although the extent of this overlap is still debated [25]. While physical pain and emotional pain differ insofar as physical pain has a sensory component (e.g. stinging, burning) [26] and emotional pain stems from psychological events rather than physical stimulation [2], they both involve an affective component (e.g. unpleasantness, distress). Prior research suggests that consoling touch reduces the affective component of physical pain [27], suggesting that consoling touch also has the potential to reduce the subjective unpleasantness associated with emotional pain.

A recent study suggests that consoling touch can increase subjective feelings of comfort and decrease responsivity in affective-pain related neural regions such as the anterior cingulate cortex and anterior insula during negative emotional experiences [28]. However, it remains an open question whether consoling touch reduces subjective feelings of emotional pain, such as the pain of social rejection, in the same way that it reduces subjective feelings of physical pain, like the pain of receiving a physical shock. One (though not the only) notable difference between emotional and physical pain is that physical pain is often temporally bound (i.e. restricted to a certain amount of time), whereas emotional pain is often more enduring. Thus, it is possible that physical and emotional pain differ in terms of how consoling touch shapes their immediate and lasting experience.

## The current investigation

This study applied a novel approach to understanding the emotional benefits of touch by examining how handholding with a romantic partner, one form of consoling touch, shapes experiences of emotional pain and comfort during emotional recollection, as well as how it shapes lasting emotional pain associated with emotional experiences. Participants recalled emotionally painful memories or neutral memories with their partners, while either holding their partner's hand or holding a squeeze-ball. Building on extensive research demonstrating the role of consoling touch in reducing physical pain, we hypothesized that handholding during the emotionally painful memories would result in lower feelings of emotional pain and greater feelings of comfort from their romantic partner as compared to holding a squeeze-ball. Since relationship satisfaction often moderates the effect of social support on wellbeing

outcomes [29–31], including the effect of handholding on the experience of physical pain [10], we additionally hypothesized that relationship satisfaction would play a moderating role in the effect of touch on emotional pain and comfort, such that greater relationship satisfaction would enhance the soothing effects of touch. Finally, participants completed an exploratory follow-up survey to assess whether there were any lasting effects of handholding on the experience of emotional pain. In other words, we aimed to test whether emotional memories paired with handholding in the lab would later be recalled with less emotional pain than those that were paired with holding a squeeze-ball. Given the exploratory nature of this follow-up study, this data is considered preliminary in elucidating how consoling touch potentially shapes the lasting experience of emotional pain.

## Methods

The University of California Los Angeles (UCLA) Institutional Review Board (IRB) has approved this study (IRB#17–001474). UCLA's Federal-Wide Assurance (FWA) with Department of Health and Human Services is FWA00004642. Informed written consent was obtained for all participants.

### Participants

We recruited 60 male-female romantic couples (*N* = 120 individuals) from the UCLA campus and surrounding community through flyers and emails. All couples were in relationships for at least 6 months (mean relationship length = 7.37 months). All interested participants separately completed an email interview to assess eligibility for participation; prospective participants who reported having any psychiatric or neurological disorder, or any serious physical illness, were not enrolled in the study. As part of this study, we also measured neural changes in participants using functional near infrared spectroscopy. These neural data were designed to serve as exploratory pilot data for a future neuroimaging study. However, because of this component of the study, all participants were required to be right-handed. The final sample (mean age = 21.81 years) included approximately 30% White, 32% Asian/Asian American, 11% Latino/a, 2% Filipino/a, 2% Black, and 11% multiracial participants. The remaining participants chose another identity or preferred not to answer.

### Sample size rationale

The rationale for our sample size derives from recently published work on affective touch [12, 32, 33]. Since these studies found effects of touch on pain with samples of 16–43 dyads, we aimed to obtain a sample of 60 dyads for our within-subjects design.

### Procedure

**Session 1.** This study included two sessions. At session 1, which took approximately 1 hour, we assigned each person within the couple to be either the "storyteller" (i.e., the person experiencing emotional pain) or the "listener" (i.e., the comforting partner). The storyteller in this study is the person who would ultimately receive support, and the listener is the person who would give support. Half of the male participants were assigned to be the person receiving support and the other half were assigned to be the person giving support so that we could examine potential gender differences in our outcome variables. Both participants completed questionnaires, and then the listener, i.e. the support giver, left the lab.

Then, the person assigned to receive support recounted stories about past experiences. For these stories, participants began by completing a form that allowed us to select which negative

stories they would recount based on: (a) whether the experience was emotionally painful at the time of the event and (b) whether they were comfortable discussing the experience on camera. This form is available on our OSF repository (titled: "Story Selection"). After selecting which stories they would share, participants recounted a total of 4–5 unrehearsed stories, each lasting about 3 minutes, as we video recorded them. Two stories were about neutral experiences, such as walking around their current residence or campus. Another 2–3 stories were about negative social experiences, such as betrayal, loneliness, loss, or rejection. We specifically asked participants not to discuss experiences that involved a former romantic partner so that they would not feel discomfort at later watching these recordings with their current romantic partner.

Participants were asked to focus their stories on how they felt at the time of the event, how they dealt with their feelings, and how they feel about the event now. To clarify, participants were not *imagining* emotional pain. Rather, they were being asked to *reflect* on and *relive* their own personal emotionally painful experiences by describing negative events from their past and the feelings those memories brought up in detail [34, 35]. This method for manipulating emotion is consistent with countless studies that have used writing or talking about past emotional experiences to evoke an emotional response [36, 37], as opposed to using impersonal standardized stimuli to induce negative affect.

During each of the recordings, the experimenter waited outside of the experiment room so that participants could recount their stories in privacy. The camera was continuously rolling throughout the session. Reminders for each prompt were presented via Qualtrics. Once the participant was ready to tell their story, they would flip a 3-minute hourglass to help them keep track of time and speak towards the camera. After each story, they responded to questions on Qualtrics about how they felt while sharing the stories (i.e. "emotional pain at first recall", see Measures for more details). At the end of the session, participants were reminded not to discuss the content of these videos with their partners until after session 2, which was approximately 1 week after session 1.

Before session 2, two experimenters independently watched the set of videos to ensure participants appropriately followed the study instructions. As they watched each video, they were asked to provide an overall rating on a scale of 1–10, 10 being the highest, based on the following question: "To what extent did the participant experience emotional pain in the video?" For videos to be considered similar enough for our experimental manipulation in session 2, ratings had to be within at least two points of each other (e.g. 9 and 7). Each rater selected the two videos that they rated as most similar based on the above question. Videos additionally had to be approximately the same length (within 18 s, about 10 seconds of the total video length). If the two raters agreed on which two videos were most similar, those videos were prepared for use in session 2. If the raters disagreed (approximately 22% of the time), a third rater was asked to provide a rating. If no consensus was reached, or no two videos approximately matched on emotionality and length, the couple was excluded from participating in session 2 (see Exclusion Criteria for details).

**Session 2.** At session 2, which took approximately 1.5 hours, the romantic couples returned to the lab together to watch 2 neutral and 2 negative videos recorded at session 1. Prior to each video, they were cued via PsychoPy to either hold hands or hold a squeeze ball such that participants underwent four conditions in a randomized order: (a) hand-holding during a negative video (i.e. consoling touch condition); (b) hand-holding during a neutral video (i.e. touch only control condition); (c) holding a squeeze ball during a negative video (emotional pain only control condition); and (d) holding a squeeze ball during a neutral video (full control condition).

After each video, participants underwent a minute of "rest" which involved closing their eyes as they continued to hold their partner's hand or hold the squeeze ball. After the rest, they

heard a beep that cued them to turn their attention to their laptops to answer questions via Qualtrics about their feelings, including "how much pain", "how hurt", "how sad", "how angry", "how much stress or anxiety", "how emotional", and "how comforted by their partner" they felt on a Likert scale of 1 to 10. The first six items were used to measure "emotional pain during the task", whereas the last item measured "comfort during the task" (see Measures for more detail). Following these questions, participants completed a shape match task for 1 minute to distract them from the previous video in preparation for watching the next video.

Throughout the task, participants sat on opposite sides of a curtain from each other to prevent them from communicating verbally or through other non-verbal cues (e.g. body language, facial expressions). Both participants could hear and see the videos on a single screen on the wall across from them, such that the support receiver and support giver experienced the stimuli simultaneously. During the two handholding conditions, they held hands through the curtain. Thus, participants were aware of the presence of their partner in all four conditions, but their contact was limited to the two conditions that included touch. Experimenters waited outside of the testing room to allow participants some privacy, but monitored the session using Google Hangouts (with participants' permission) to ensure that the task ran smoothly and that participants followed instructions. At the end of all four story-videos, participants completed a brief end-of-study questionnaire.

**Exclusion criteria.** Participants were not invited to return to the lab for session 2 if the videos they recorded at session 1 were unusable, either because of technological issues with the recordings or because they did not follow the video prompt instructions. Ten couples were not eligible for session 2 for this reason. Two additional couples dropped out of the study after session 1 due to scheduling issues. One additional couple was removed from analyses due to technological issues during session 2, leaving a total of 47 couples in the sample.

**Follow-up survey.** To examine potential lasting effects of consoling touch on emotional pain, participants who received support completed a brief exploratory follow-up survey. These surveys were sent out electronically after we completed in-lab data collection for all of our dyads. Thus, participants completed the survey between 1.28 and 7.82 months after session 2 ($M$ = 4.01 months). Because we decided to add this follow-up assessment to our investigation while data collection was ongoing, this portion of the project was an exploratory addition to the original research plan. These surveys included personalized cues to remind participants of the emotional stories they shared (e.g. "When you participated in our study you recorded a video about the loss of your grandmother."), followed by questions about how much emotional pain they associated with those memories, including how much pain they experienced at the time of the event, and how much pain they experience when thinking about the event now (i.e. lasting emotional pain, see Measures for more detail). We received responses from 79% of our couples ($N$ = 31).

## Measures

**Emotional pain at first recall.** To account for the fact that different memories can elicit differing amounts of emotional pain, we asked participants to rate the extent to which they felt different negative emotions on a scale of 1 to 10 during each video recording at session 1. These negative emotion ratings included "how much pain", "how hurt", "how sad", "how angry", "how much stress or anxiety", and "how emotional" they felt. Cronbach's alpha for these items on conditions involving emotional content (negative video 1: $\alpha$ = 0.90; negative video 2: $\alpha$ = 0.91) indicated a high degree of covariance between the six individual items. Thus, the items were averaged into a single "emotional pain at first recall" composite for each of the

stories that we used to control for differences in the relative painfulness of reliving the different memories.

**Emotional pain and comfort during the task.** To assess how participants felt when recalling the memories with their partner, and either holding hands or holding a squeeze-ball, we asked them the same set of negative emotion questions used to create the "emotional pain at first recall" composite described above. Once again, Cronbach's alpha indicated a high degree of covariance between the six individual negative affect items (consoling touch condition: α = 0.92; emotional pain only condition: α = 0.92). Thus, the items were averaged into a single "emotional pain during the task" composite for each condition.

Meanwhile, "comfort during the task" was assessed with a single item asking participants how comforted they felt by their romantic partners as they recalled each memory on a scale of 1 to 10. To test the association between participants' emotional pain and comfort, we examined the correlation between these variables in each condition. These variables were correlated during the consoling touch condition, $r = 0.47$, $p < 0.01$, and the emotional pain only condition, $r = 0.30$, $p = 0.04$, but not during the touch only condition (i.e. holding hands during a neutral video), $r = 0.14$, $p = 0.37$, or the full control condition (i.e. holding a squeeze-ball during a neutral video), $r = 0.03$, $p = 0.83$. Since emotional pain and comfort represent related but distinct concepts, they were maintained as separate outcome variables.

**Lasting emotional pain.** To assess whether touch shaped the lasting experience of emotional pain, we asked participants the following questions in a follow-up survey on a Likert scale of 1 to 10: (a) "How much pain did you experience at the time of the event that you described in that video?" and (b) "When you think about this experience now, how much pain do you experience?" The first question would serve as a baseline assessment of how participants recalled feeling at the time that the event originally occurred, and the second question would indicate how much emotional pain they currently associated with the event (i.e. lasting emotional pain).

**Relationship satisfaction.** As a measure of relationship satisfaction, participants completed the 32-item version of the Couples Satisfaction Index (CSI), which includes items such as "I have a warm and comfortable relationship with my partner" and "I really feel like part of a team with my partner" [38]. The full scale can be accessed through our OSF repository.

## Statistical analyses

For our analyses, we used the statistical package R (Version 1.2.1335) to create linear mixed models (LMMs, i.e. multilevel regression) with participant ID as the group level variable, fixed effects, and random intercepts. We used the "lmer" package in R, which by default uses the Satterthwaite degrees of freedom method and bases confidence intervals and $p$-values on the $t$-distribution. This analytic approach allowed us to account for non-independence of errors due to our repeated-measures design, which would result in underestimated standard errors and inflated risk of type I error, while also providing more modeling flexibility than repeated-measures ANOVA. Since repeated-measures ANOVA only uses list-wise deletion, multilevel regression is additionally better at accounting for missing data, and therefore has greater statistical power than repeated-measures ANOVA. Data and analysis materials can be accessed through Open Science Framework upon request to the first author.

As a first step, we conducted exploratory analyses to investigate potential gender differences in our primary outcome variables (i.e. emotional pain, comfort) by running LMMs with valence of the videos, touch, gender, and interactions between them as predictor variables. There were no significant main effects or interactions with gender on any of the outcome variables ($p$'s > 0.05). Thus, gender was not included in any subsequent models. A full report of these analyses is included in our S1 File.

To test how touch shaped emotional outcomes for participants as they recalled the memories with their partners, we ran two separate LMMs with emotional pain during the task and how comforted they felt by their partner as the outcome variables. These models included valence of the videos (negative vs. neutral), touch (hand vs. ball), and the interaction between them as predictors. To account for potential differences in the emotional intensity of the different memories being recalled, we included the measure of emotional pain at first recall (assessed at session 1) as a covariate in these analyses.

To follow-up on these analyses, we conducted tukey-adjusted pairwise comparisons with a focus on the contrast between the consoling touch condition and the emotional pain only control condition (i.e. holding a squeeze-ball during a negative video) to examine the following hypotheses: (1) participants' emotional pain will be significantly lower during consoling touch than emotional pain only, and (2) participants' comfort will be significantly higher during consoling touch than emotional pain only.

Both of these models (assessing the effect of consoling touch on participants' emotional pain and comfort) were then re-run with participants' relationship satisfaction (i.e. CSI) as a possible moderator of the association between consoling touch and emotional pain/comfort. These models included valence, touch, relationship satisfaction, and interactions between them as predictors, and emotional pain at first recall as a covariate. We followed up on significant interaction terms that included relationship satisfaction by obtaining estimated marginal means for our model using the "emmeans" package. This method uses the given model to approximate the outcome variable at different levels of a continuous moderator, adjusting for other variables in a model [39].

To analyze the results of our follow-up survey probing potential lasting effects of consoling touch, we ran a separate LMM with touch as the predictor and participants' current emotional pain when thinking about the event as the outcome variable. To clarify, valence was not included in this model because the follow-up survey only asked about the two negative emotional memories, not the two neutral memories. To account for potential differences in the emotional intensity of the different memories being recalled, we included how participants recalled feeling at the time of the event (assessed at follow-up) as a covariate in this analysis. We then ran this analysis with relationship satisfaction as a possible moderator of the effect of touch on current emotional pain. To assess whether the amount of time between the in-lab session and the follow-up survey affected the results, we also ran the model with time as an additional control variable. Finally, since this follow-up study involved a smaller subset of participants than our primary analyses, we re-ran our primary analyses using this smaller subset of participants to ensure consistency in our results (see S1 File for a full report of these analyses and accompanying figures).

## Results

### Does consoling touch decrease emotional pain?

Controlling for potential differences in the emotional intensity of the different memories being recalled (i.e. emotional pain at first recall), $b = 0.49$, $t(175.83) = 9.96$, $p < .001$, 95% CI = [0.39, 0.59], there was a significant main effect of valence, $b = -1.18$, $t(170.47) = -4.47$, $p < .001$, 95% CI = [-1.69, -0.66], no main effect of touch, $b = -0.05$, $t(136.18) = -0.28$, $p = 0.78$, 95% CI = [-0.44, 0.33], and no interaction between valence and touch, $b = 0.05$, $t(135.50) = 0.16$, $p = 0.87$, 95% CI = [-0.50, 0.59], on how much emotional pain participants felt. Participants felt significantly more emotional pain during the negative videos ($M = 4.17$, $SD = 1.82$) than the neutral videos ($M = 1.23$, $SD = 0.35$) (Fig 1A). Contrary to our hypothesis, pairwise comparisons indicated no significant difference between how much emotional pain participants

felt during the consoling touch condition versus the emotional pain only condition, $t(136) =$ 0.28, $p = 0.99$, 95% CI = [-0.46, 0.57]. When including participants' relationship satisfaction scores in the model as a potential moderator, we found that relationship satisfaction did not moderate the effects of touch, $b = 0.01$, $t(132.48) = 1.17$, $p = 0.24$, 95% CI = [-0.01, 0.04], valence, $b = 0.00$, $t(132.48) = 0.37$, $p = 0.71$, 95% CI = [-0.02, 0.03], or the interaction between touch and valence on participants' emotional pain, $b = -0.02$, $t(132.45) = -1.03$, $p = 0.31$, 95% CI = [-0.05, 0.01].

### Does consoling touch increase feelings of comfort from one's partner?

Controlling for potential differences in the emotional intensity of the different memories being recalled (i.e. emotional pain at first recall), $b = 0.22$, $t(177) = 2.09$, $p = 0.04$, 95% CI = [0.01, 0.44], there was no main effect of valence, $b = 0.12$, $t(160.71) = 0.22$, $p = 0.83$, 95% CI = [-0.98, 1.23], a significant main effect of touch, $b = 2.80$, $t(131.95) = 6.74$, $p < .001$, 95% CI = [1.99, 3.61], and no interaction between valence and touch, $b = -0.84$, $t(131.35) = -1.43$, $p = 0.15$, 95% CI = [-1.98, 0.30], on how comforted participants felt by their partner. Participants felt more comforted by holding their partners' hand ($M = 5.2$, $SD = 2.82$) than by holding a squeeze ball ($M = 2.77$, $SD = 2.16$) (Fig 1B). Consistent with our hypothesis, pairwise comparisons indicated that participants felt significantly more comforted during the consoling touch condition ($M = 5.98$, $SD = 2.59$) than the emotional pain only condition ($M = 3.11$, $SD = 2.01$), $t(133) = -6.74$, $p < .001$, 95% CI = [-3.88, -1.72].

When including participants' relationship satisfaction scores in the model as a potential moderator, we found that relationship satisfaction did not moderate the association between valence by touch and comfort, $b = -0.05$, $t(128.22) = -1.42$, $p = 0.16$, 95% CI = [-0.11, 0.02], or the association between valence and comfort, $b = 0.00$, $t(128.23) = 0.16$, $p = 0.88$, 95% CI = [-0.04, 0.05], but did significantly moderate the association between touch and comfort, $b = 0.07$, $t(128.27) = 2.96$, $p = .004$, 95% CI = [0.02, 0.11]. While relationship satisfaction did not enhance comfort during the squeeze-ball conditions, $b = 0.02$, $t(74.4) = 0.04$, $p = 0.97$, it did enhance comfort during handholding such that those with high relationship satisfaction

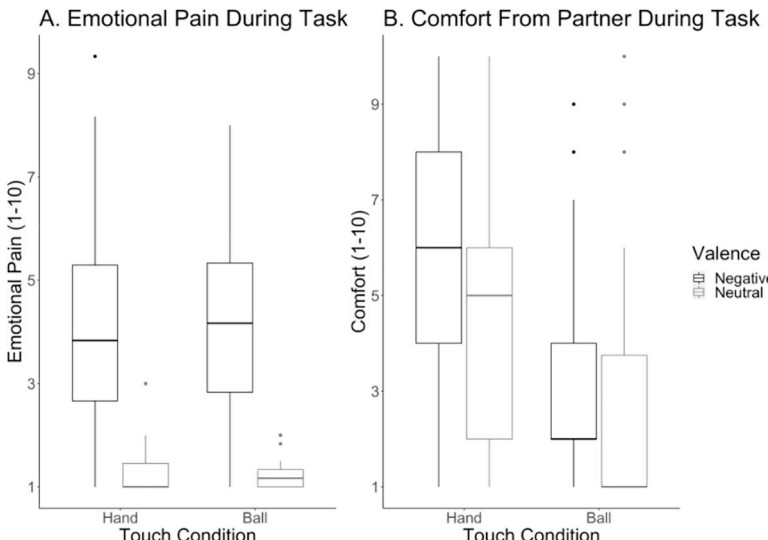

**Fig 1. Handholding does not decrease emotional pain, but does increase comfort.** Participants' feelings of emotional pain (A) and comfort (B) plotted by touch and valence conditions. Participants felt more emotional pain during the negative videos than the neutral videos, and felt more comforted when handholding during the negative video (i.e. consoling touch) than holding a squeeze-ball during the negative video.

reported greater comfort during handholding (*EMM* = 5.97) as compared to those with low relationship satisfaction (*EMM* = 4.39), *b* = 1.59, *t*(74.6) = 2.82, *p* = .006 (Fig 2).

## Is there a lasting effect of consoling touch on how emotionally painful events are recalled?

When controlling for emotional pain at the time of the original emotional event ("How much pain did you experience at the time of the event that you described in that video?"), *b* = 0.50, *t*(54.03) = 5.37, *p* < .001, 95% CI = [0.32, 0.69], touch significantly predicted current pain about the emotional memory ("When you think about this experience now, how much pain do you experience?"), *b* = -0.58, *t*(29.02) = -2.24, *p* = 0.03, 95% CI = [-1.09, -0.07]. Specifically, participants' current pain when recalling the past event was lower for the emotionally painful memory that was previously paired with handholding (*M* = 3.74, *SD* = 2.08) as opposed to the emotionally painful memory that was previously paired with holding a squeeze ball (*M* = 4.26, *SD* = 1.88) (Fig 3). When including relationship satisfaction as a moderator in the model, we did not find a significant interaction between touch and relationship satisfaction, *b* = -0.02, *t* (28.06) = -0.75, *p* = 0.46, 95% CI = [-0.06, 0.02].

   In addition, when controlling for emotional pain at the time of the original emotional event, *b* = 0.50, *t*(53.38) = 5.33, *p* < .001, 95% CI = [0.32, 0.69], *and* the amount of time that passed between the in-lab manipulation and the completion of the follow-up survey, *b* = -0.00, *t*(28.35) = -0.78, *p* = 0.44, 95% CI = [-0.01, 0.00], we still found a significant effect of touch on current emotional pain, *b* = -0.58, *t*(29.05) = -2.24, *p* = 0.03, 95% CI = [-1.09, -0.07]. Furthermore, when using our measure of emotional intensity from the in-lab portion of the study (i.e. emotional pain at first recall), *b* = 0.44, *t*(57.32) = 3.81, *p* < .001, 95% CI = [0.21, 0.66], instead of participants' self-reported emotional pain at the time of the original event assessed at follow-up, we still found a significant effect of touch on current emotional pain, *b* = -0.59, *t* (29.76) = -2.15, *p* = 0.04 95% CI = [-1.12, -0.05].

## Discussion

A robust body of work demonstrates that consoling touch can reduce the affective experience of physical pain [10–12], and that physical pain and emotional pain share a common neural system [22–24, c.f. 25]. Intuitively, then, we may assume that consoling touch reduces subjective reports of emotional pain. Surprisingly, however, our results indicate that consoling touch does not decrease the immediate subjective experience of emotional pain relative to holding a squeeze ball in the presence of one's romantic partner. But, it does lead individuals to feel more comforted by their partner than holding a squeeze ball, particularly when they have greater relationship satisfaction with their partner. This finding is in line with other work suggesting that consoling touch increases subjective feelings of comfort during emotional pain [28], and that relationship satisfaction plays an important role in how we feel during consoling touch [10]. We found no effect of gender on our outcome variables. Given that the majority of touch studies have only examined female participants [10, 12, 21], with some work finding gender differences during the experience of physical pain [20], further research is needed to clarify how gender shapes the outcome of consoling touch.

   A somewhat surprising finding was that participants' feelings of emotional pain and comfort were *positively* correlated in the emotional conditions, particularly during consoling touch. Intuitively, comfort and emotional pain seem to be opposite experiences, such that feeling comforted by a social support figure should result in lower subjective emotional pain. However, participants in our study reported feeling more comfort from their partners as they experienced greater emotional pain, suggesting that emotional pain may be to some extent

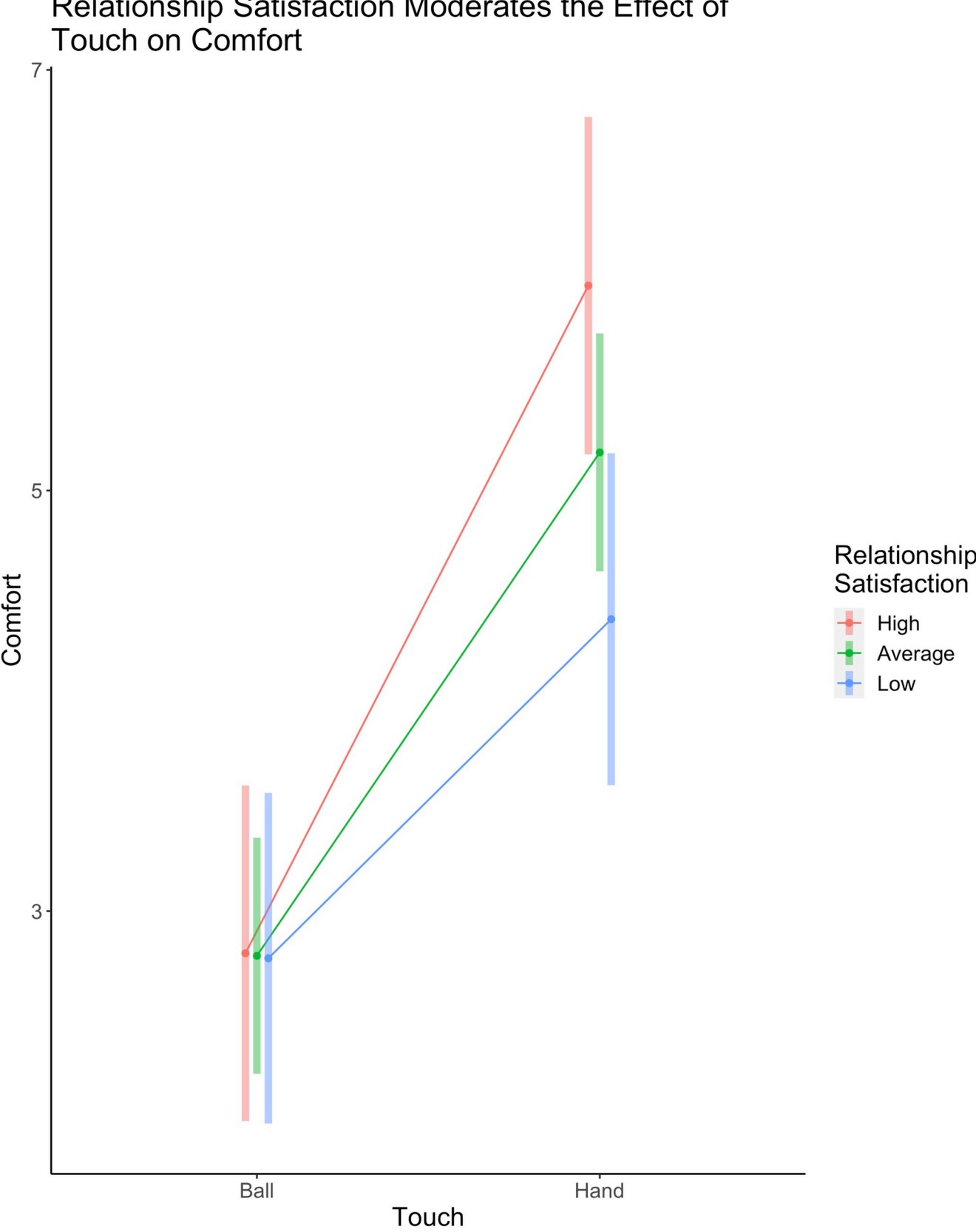

**Fig 2. Relationship satisfaction moderates the effect of handholding on feelings of comfort.** While relationship satisfaction did not enhance comfort during the squeeze-ball conditions, it did enhance comfort during handholding such that those with high relationship satisfaction had greater comfort during handholding as compared to those with low relationship satisfaction.

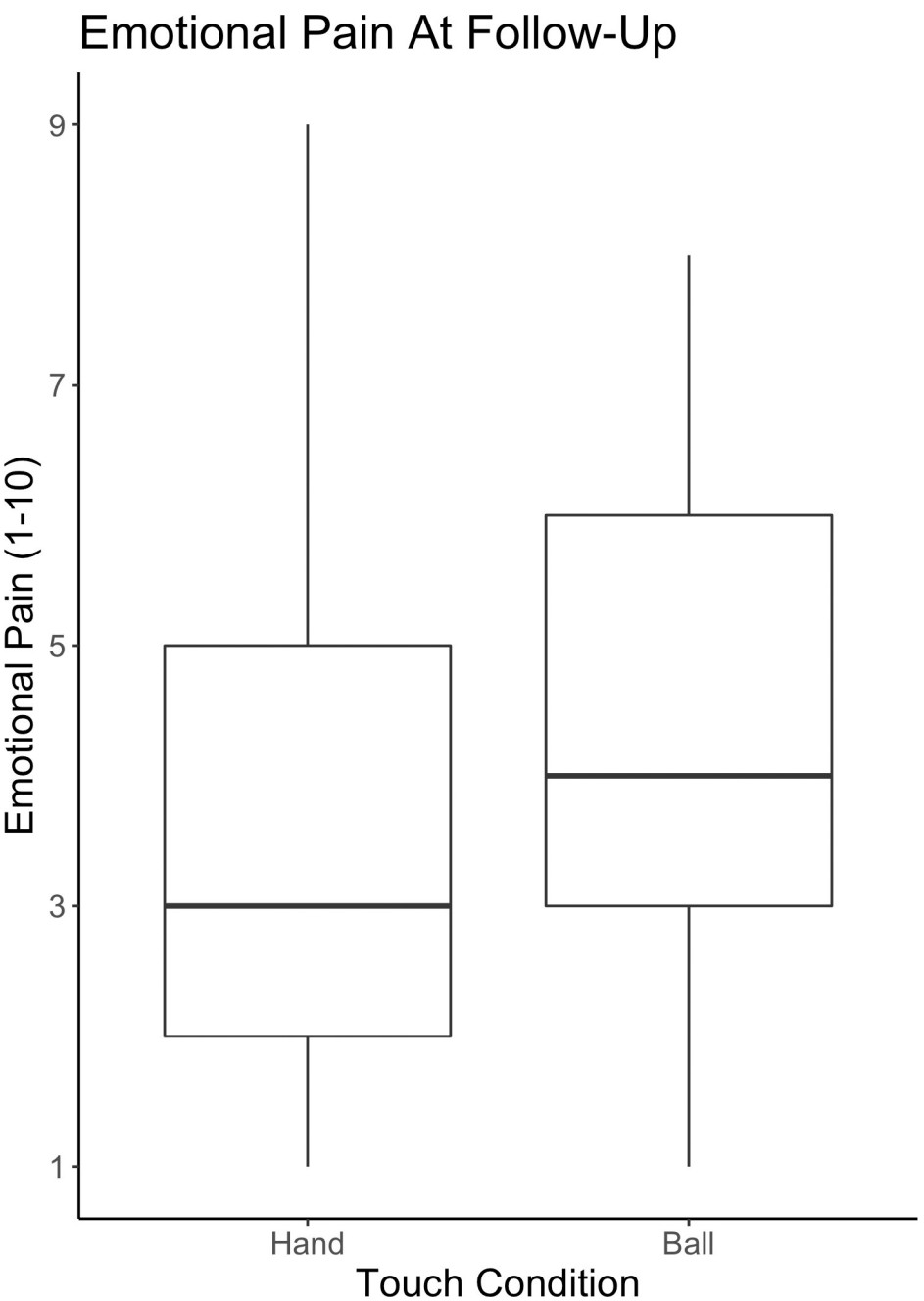

**Fig 3. Negative memories that are associated with handholding in the lab are later recalled with lower emotional pain.** Emotional pain when recalling a negative emotional memory at follow-up plotted by touch condition (hand or ball). Participants felt significantly less pain recalling an emotional memory after that memory was paired in the lab with handholding as compared to holding a squeeze ball.

necessary to feel comforted. In other words, people may feel the most comforted when such comfort is needed (i.e. there is something to comfort them about). This finding is interesting to consider in terms of how negative emotional experiences are socially regulated, and whether feelings of comfort simply provide a source of positive affect during a negative emotional experience, as opposed to decreasing the negative affect associated with that experience.

This set of findings suggests a potentially important difference between emotional pain and physical pain in terms of how negative experiences are regulated in these two contexts. During physical pain, it may be adaptive to down-regulate immediate distress, particularly in a lab setting where the pain is not necessarily helpful in recognizing and escaping some sort of threat. However, during emotional pain, down-regulating immediate distress may not always be adaptive since such distress may be stemming from personally meaningful events that need to be processed and reflected on over time. Indeed, research suggests that individuals often feel motivated to experience negative emotional states because they are helpful in navigating certain experiences (e.g. anger when preparing for a conflict, sadness in coping with a loss), even if those emotional states feel unpleasant [40–42]. Thus, subjective distress may be necessary to some extent to process emotional memories in a way that supports adaptive long-term outcomes and resolution. This idea is consistent with research demonstrating that exposure to negative emotional stimuli (e.g. spiders for phobic patients) can be instrumental in reflecting on and changing harmful cognitions associated with those stimuli over time [43, 44]. In other words, we may need to feel certain emotions in order to process and learn from them, allowing us to heal and regulate over time.

Although preliminary, a particularly interesting finding is that emotional memories that were paired with touch were later recalled with less emotional pain than those paired with holding a squeeze-ball. While this finding is novel with regards to the effect of touch on emotional memories, it is interesting in light of other research showing that close others can activate feelings of safety, which can have lasting effects on fear-learning processes [45, 46]. Thus, it is possible that consoling touch promotes a feeling of safety while recalling emotionally painful memories, facilitating a form of counter-conditioning that diminishes the negative affect associated with the painful memory [47].

Ultimately, this study indicates an interesting distinction between the social regulation of physical and emotional pain via touch, and suggests that while consoling touch can be helpful in both contexts, it may be helpful in different ways. Although this study sheds light on the potential benefits of consoling touch in emotional contexts, it is only a preliminary step in understanding how consoling touch supports emotional well-being. The potential lasting effect of consoling touch on the experience of emotional memories particularly warrants future investigation, and boundaries on this effect should be explored in terms of when and how changes in emotional pain take place, and whether the experience of such emotional memories is related to other measures of well-being. Additionally, future work can build on these findings to more specifically target certain mechanisms that might predict the magnitude of these effects, and explain possible pathways through which consoling touch shapes the immediate and lasting experience of emotional pain.

## Supporting information

**S1 File.**
(DOCX)

## Acknowledgments

We are thankful for thoughtful feedback on study design and results from members of the UCLA Social and Affective Neuroscience Lab and UCLA Social Cognitive Neuroscience Lab.

## Author Contributions

**Conceptualization:** Razia S. Sahi, Macrina C. Dieffenbach, Shannon M. Burns, Matthew D. Lieberman, Naomi I. Eisenberger.

**Data curation:** Razia S. Sahi, Macrina C. Dieffenbach, Siyan Gan, Maya Lee, Laura I. Hazlett.

**Formal analysis:** Razia S. Sahi, Macrina C. Dieffenbach.

**Funding acquisition:** Simone G. Shamay-Tsoory, Naomi I. Eisenberger.

**Investigation:** Razia S. Sahi, Naomi I. Eisenberger.

**Methodology:** Razia S. Sahi, Naomi I. Eisenberger.

**Project administration:** Siyan Gan, Maya Lee.

**Resources:** Shannon M. Burns, Matthew D. Lieberman, Naomi I. Eisenberger.

**Software:** Shannon M. Burns.

**Supervision:** Macrina C. Dieffenbach, Naomi I. Eisenberger.

**Writing – original draft:** Razia S. Sahi, Naomi I. Eisenberger.

**Writing – review & editing:** Razia S. Sahi, Macrina C. Dieffenbach, Shannon M. Burns, Matthew D. Lieberman, Simone G. Shamay-Tsoory, Naomi I. Eisenberger.

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
