## [Decision Letter · Decision Letter 0]

29 Sep 2020

PONE-D-20-25552

The Comfort in Touch: Immediate and Lasting Effects of Handholding on Emotional Pain

PLOS ONE

Dear Dr. Sahi,

Thank you for submitting your manuscript to PLOS ONE. After careful consideration, we feel that it has merit but does not fully meet PLOS ONE’s publication criteria as it currently stands. Therefore, we invite you to submit a revised version of the manuscript that addresses the points raised during the review process.

Specifically, all three reviewers point out that the framing of the study could be better linked to the actual study, in terms of that targeting memory related pain instead of acute pain (see reviewers 3&2), and it could be sharpened by more specificity in the data analysis and interpretation based on potential confounds (see reviewer 1).

We look forward to receiving your revised manuscript.

Kind regards,

Hedwig Eisenbarth

Academic Editor

PLOS ONE

Journal Requirements:

2. Please change "Caucasian” to “White” or “of [Western] European descent” (as appropriate).

Reviewers' comments:

Reviewer's Responses to Questions

**Comments to the Author**

1. Is the manuscript technically sound, and do the data support the conclusions?

Reviewer #1: Partly

Reviewer #2: Partly

Reviewer #3: Yes

2. Has the statistical analysis been performed appropriately and rigorously? 

Reviewer #1: Yes

Reviewer #2: No

Reviewer #3: Yes

3. Have the authors made all data underlying the findings in their manuscript fully available?

Reviewer #1: Yes

Reviewer #2: Yes

Reviewer #3: Yes

4. Is the manuscript presented in an intelligible fashion and written in standard English?

Reviewer #1: Yes

Reviewer #2: Yes

Reviewer #3: Yes

5. Review Comments to the Author

Reviewer #1: This study investigates the influence of handholding on emotional distress and emotional comfort to recollection of emotionally valanced memories. The during-task analysis consisted of 47 heterosexual couples and 31 of those completed the post-task follow-up tasks for that analysis. The authors show that participants reported handholding increasing their feelings comfort during the task but not reducing distress. Additionally, the authors indicate that the follow-up analysis findings reveal that participants’ later recalled memories of the experiment were less painful during the handholding condition than the control condition.

The research questions of the study have merit and I was enthusiastic to read the findings from this study. I found the research methods—including data collection procedures and statistical analyses—to be appropriate for the research questions. However, I do have some major and minor concerns (each of these are listed below) which I feel impact the validity of the findings. Overall, I think that there are some issues that confound the interpretation of the findings and unless some follow-up studies or extra data collection are not performed, I am not confident these findings would hold.

Major Concerns Overview:

I am not convinced of the findings from the follow-up analyses. These results seem to be at odds with the results obtained from the experimental portion. The data from the experimental portion is much cleaner and less prone to confounds. The follow-up data is a single question rather than a more robust scale. The time frame spanned an average of 4 months. The way in which the authors determined “baseline pain” is confounded because they are asking participants to recall their baseline (rather than actually measuring it at baseline). Additionally, by this point the sample size is a mere 31 dyads.

The relationship quality measure and analyses are so lacking in detail that it can’t be interpreted as is. Potentially this can be resolved simply by more details in the manuscript. However, as it is reported in the manuscript now, it is not appropriate for the analyses conducted.

Because of these confounds, this paper is essentially contributing 1) the experimental manipulation of negative memories was more distressing than neutral memories—no condition (handholding) differences, 2) participants felt more comfort from spousal handholding than from control (squeeze-ball) during both negative and emotional memories. Neither of these findings are particularly noteworthy nor do they meaningfully contribute to the literature.

Major Concerns Manuscript Body:

The introduction has some holes as it relates to the research questions investigated. One of the primary stated hypotheses has to do with relationship quality, however, I was greatly wanting for any literature that discusses this topic. The only info listed is a study from nearly 15 years ago... This is not sufficient (other intro holes are addressed in the minor concerns below).

Your sample size was vastly shrinking by the end of the study. It started out at a decently sized 60 dyads, then down to 47 for the session one and two analyses and finally only 31 dyads for the follow-up survey analyses. That is a pretty small sample size for the follow-up analyses, which is where the contribution of this paper hinges. Because of this, you are no longer examining your primary, controlled experiment data. These follow-up analyses would require more power to detect true differences because of the time elapsed and the self-report recollection nature of the reporting. This is a major confound in your results.

Rather than the current rationalization for your sample size (and you can’t really conduct a power analysis to determine your sample size, since that is done), I suggest you run and report an analysis of how likely it was that you observed a significant effect, given your sample, and given your expected effect sizes.

Continuing on with the follow-up analysis. It could be a pretty big confound to ask people to retrospectively rate how they felt and then at the same time as asking them how they felt now. This could greatly skew your baseline. Why not use the data you already have from them for when they reported how they felt during the experiment as the baseline rather than forcing them to recall how they felt and then imposing that as a baseline?

A major concern is the way the relationship quality measure is used (potentially just the way you report it?). In the intro, measures, and in the results sections, info is severely lacking. In the measures you only state two items. Was the full scale not used? If not, your relationship quality analyses are greatly confounded; what is the point of having the scale if you only used two items from it? If so, these sections need to be clarified. What are your reliability stats, means, SD, etc. Additionally, how did you use the scale in your analyses? From the figure it looks like you classified couples into high, med, and low RQ… But HOW? How many of your participant-couples fit into each category? Is that same classification used for all your analyses? Additionally, the “Funk measure’s” name is the “Couple Satisfaction Index” or CSI.

Minor Concerns:

Your research paradigm could use some bolstering in how imagining experiences or memories is an appropriate research method. I am not saying it is not appropriate. I am just recommending you address this and justify it with relevant literature. I have listed some suggested references at the bottom that you may want to look into.

You address how “it is unclear whether touch reduces emotional pain the same as physical pain”. But you never detail why emotional pain matters. You should provide more background and justification on why emotional pain is important and a relevant construct separate from and beyond physical pain. Why should readers care about emotional pain? Addressing this will give the readers a clearer view of why this study matters.

Switch the order in presentation of method section. Logically before participants tell about their stories, we should be told how they chose which stories to tell.

How was the prompt on the computer monitor programmed/controlled? Did you use e-prime or some other software that ran automatically? Did you have RA’s running it in the background?

p.7 you describe how participants fill out a form which RA’s then selected from to determine which story they would tell. More detail on this form should be included. Possibly just including the form as supplementary material.

p.8, first paragraph: What is meant by “After each story, they responded to questions on the computer monitor about how they felt while sharing the stories.”? Is this part of questions reported in your measures? If so, which measure are you referring to? Or are these questions something different? If so, what does “how they felt” mean or entail? How they felt is too vague.

p.8 “two experimenters independently watched the set of videos… and selected two negative videos for viewing in session two”. Were there specific coding by the RA’s to determine if they were emotional? What were the criteria the RA’s used for this decision? Was there any interrater reliability scores you can report? You should have some sort of justification on how their subjective opinion on “emotional” be validated or justified?

Measures section, p. 11 you use the term “’relationship’ between participants’ distress and comfort”. This term should be ‘relation’. Relationship means that thing between real people. Inanimate objects have a ‘relation’ not a ‘relationship’. Especially since you have “relationship quality” as a primary term in your study. That could get confusing.

I am guessing that your Comfort scale is on the same 1-10 scale as your distress scales was? This should be clarified in that section.

Suggested literature:

Jakubiak, B. K., & Feeney, B. C. (2016). Keep in touch: The effects of imagined touch support on stress and exploration. Journal of Experimental Social Psychology, 65, 59-67. doi:https://doi.org/10.1016/j.jesp.2016.04.001

Graff, T. C., Luke, S. G., & Birmingham, W. C. (2019). Supportive hand-holding attenuates pupillary responses to stress in adult couples. PLOS One, 14(2), e0212703. doi:10.1371/journal.pone.0212703

Jakubiak, B. K., & Feeney, B. C. (2017). Affectionate Touch to Promote Relational, Psychological, and Physical Well-Being in Adulthood: A Theoretical Model and Review of the Research. Personality and Social Psychology Review, 21(3), 228–252. https://doi.org/10.1177/1088868316650307

Reviewer #2: Please see attached comments.

Reviewer #3: The manuscript is well-written and provides novel insights on the effect of touch on distress and comfort and lasting pain during emotional recollection. I have several issues that should be adressed

Introduction

Generally, the introduction mostly sounds reasonable. However, the manuscript deals with memory-related pain/distress but this issue was not connected to the proposed rational.

Methods:

The description of the analytical approach is very general and not specific to the current study.

For example, what residual distribution was assumed? What degrees of freedom were used?

What random effects/covariance structures were assumed?

Please, provide a detailed description of the analysis regarding the gender effect

6. PLOS authors have the option to publish the peer review history of their article (what does this mean?). If published, this will include your full peer review and any attached files.

Reviewer #1: No

Reviewer #2: No

Reviewer #3: No

---

## [Author Response · Author response to Decision Letter 0]

24 Dec 2020

Round 1 Response to Editor and Reviewers

Note to Editor and all Reviewers: In line with recommendations for enhanced clarity, we have changed the naming of our variables as follows: (1) instead of “Memory-Related Distress”, we now use “Emotional Pain at First Recall”; (2) instead of “Task-Related Distress”, we now use “Emotional Pain During the Task”; (3) instead of “Task-Related Comfort”, we now use “Comfort During the Task.” Instead of using both “distress” and “emotional pain” in various places in the manuscript, we now consistently use the term “emotional pain” and offer a concise definition of this concept on page 3 of the manuscript. Thank you for your recommendations to improve our manuscript. 

Editor Comments

Comment 1: Thank you for submitting your manuscript to PLOS ONE. After careful consideration, we feel that it has merit but does not fully meet PLOS ONE’s publication criteria as it currently stands. Therefore, we invite you to submit a revised version of the manuscript that addresses the points raised during the review process. Specifically, all three reviewers point out that the framing of the study could be better linked to the actual study, in terms of that targeting memory related pain instead of acute pain (see reviewers 3&2), and it could be sharpened by more specificity in the data analysis and interpretation based on potential confounds (see reviewer 1).

Response 1: We thank you and the reviewers for your thoughtful feedback and the opportunity to improve upon our manuscript. We have responded to each comment provided by you and the reviewers in what follows. We hope you’ll agree that this revised manuscript is now ready to share with the scientific community. 

To summarize, we have made the following major revisions:

- We have added greater detail to our introduction to clarify our focus on emotional pain, specifically discussing why it is important to study and how this study meaningfully extends the existing literature on consoling touch and physical pain. 

- We have added more information regarding how we’ve operationalized emotional pain, and why recalling painful memories is not an example of imagined pain, but rather a way to reflect on and relive one’s own personal emotional pain (providing several related citations). 

- We have added more information regarding our data analysis and results, including our gender differences analyses, relationship satisfaction moderation analyses, and different possible covariates for the follow-up analyses.

- We have revised some of our language to be more consistent throughout, and updated the naming of our variables for increased clarity (see “Note to Editor and all Reviewers”). 

- Additional materials have been added to our OSF repository to enhance the reproducibility of our work, including details of each of our measures, our story selection form (referenced on page 8), our protocols for each session, and the variables used to create our composite variables (see dataset). As a reminder, we do not have IRB permission to share the data publically, so it is only available to view through a private link: https://osf.io/9wbua/?view_only=de32be8c75a94f1e9d3bdb9c7c21f19f. 

Comment 2: If applicable, we recommend that you deposit your laboratory protocols in protocols.io to enhance the reproducibility of your results.

Response 2: We have added our protocols to the OSF repository for this study in order to maintain a single location where all of our study materials are located.

Comment 3: Please ensure that your manuscript meets PLOS ONE's style requirements, including those for file naming.

Response 3: Our manuscript and file names have been revised to meet PLOS ONE’s style requirements.

Comment 4: Please change "Caucasian” to “White” or “of [Western] European descent” (as appropriate).

Response 4: We have changed “Caucasian” to “White” on page 6.

Reviewer #1:

Comment 1: The research questions of the study have merit and I was enthusiastic to read the findings from this study. I found the research methods—including data collection procedures and statistical analyses—to be appropriate for the research questions. However, I do have some major and minor concerns (each of these are listed below) which I feel impact the validity of the findings. Overall, I think that there are some issues that confound the interpretation of the findings and unless some follow-up studies or extra data collection are not performed, I am not confident these findings would hold.

Response 1: We thank you for your feedback and the suggestions for improving upon our manuscript. We have taken steps to address your concerns, detailed below. While additional data collection to follow-up on our results is currently not possible due to the ongoing pandemic, nor did the Editor or other Reviewers request it, we agree that future work can and should build on the findings reported in this paper to further understand this phenomenon. Given the current situation as well as our responses to your concerns below, we hope that you will agree with us that it is better to publish these findings (with limitations clearly stated) to get them out to the scientific community rather than wait possibly multiple years to follow up on them. 

Comment 2: I am not convinced of the findings from the follow-up analyses. These results seem to be at odds with the results obtained from the experimental portion. The data from the experimental portion is much cleaner and less prone to confounds. The follow-up data is a single question rather than a more robust scale. The time frame spanned an average of 4 months. The way in which the authors determined “baseline pain” is confounded because they are asking participants to recall their baseline (rather than actually measuring it at baseline). Additionally, by this point the sample size is a mere 31 dyads.

Response 2: Thank you for your thoughtful reflection on the follow-up study. While the results from the follow-up study differ from the results obtained in the experimental portion, we do not believe these findings are necessarily at odds with each other, but rather that they may collectively paint a broader picture of the underlying phenomenon. Previous research has demonstrated that emotional experiences can be regulated over time such that some interventions do not confer immediate results, but benefit individuals over time. For example, affect labeling is a form of emotion regulation that sometimes confers no immediate benefits, but considerably benefits patients with phobia over time (i.e. lower skin conductance response relative to other groups from immediate posttest to 1-week posttest, but not from pretest to immediate posttest; Kircanski et al., 2012).

We agree that it is not ideal that the time frame between the in-lab session and follow-up survey varied so much across participants. However, we did rerun our analyses of the follow-up study controlling for the amount of time that passed between the in-lab manipulation and the completion of the follow-up survey and the results held. Specifically, our finding regarding the effect of touch on current emotional pain was consistent across these analyses: when controlling for pain at the time of the original emotional event, touch significantly affected current emotional pain, regardless of whether we controlled for time between the in-lab study and follow-up survey. This provides us with some evidence that the time frame was not a major confound in this analysis. We have added additional details to describe these analyses on page 19:

“In addition, when controlling for emotional pain at the time of the original emotional event, b = 0.50, t(53.38) = 5.33, p < .001, 95% CI =[0.32, 0.69], and the amount of time that passed between the in-lab manipulation and the completion of the follow-up survey, b = -0.00, t(28.35) = -0.78, p = 0.44, 95% CI =[-0.01, 0.00], we still found a significant effect of touch on current emotional pain, b = -0.58, t(29.05) = -2.24, p = 0.03, 95% CI =[-1.09, -0.07].”

To address your concern about the baseline pain measure utilized in the follow-up analyses as a covariate, we re-ran our analyses for this follow-up study using the same baseline as we used in our main analyses (emotional pain at first recall which is a composite of 6 items rather than a single item). We found the same pattern of results when using our baseline measure of emotional pain from the experimental study as we do when using the one item baseline measure from the follow-up study. We have added a description of these analyses on page 19:

“Furthermore, when using our measure of emotional intensity from the in-lab portion of the study (i.e. emotional pain at first recall), b = 0.44, t(57.32) = 3.81, p < .001, 95% CI =[0.21, 0.66], instead of participants’ self-reported emotional pain at the time of the original event assessed at follow-up, we still found a significant effect of touch on current emotional pain, b = -0.59, t(29.76) = -2.15, p = 0.04 95% CI =[-1.12, -0.05].”

We agree that the experimental portion of this project is more robust than the follow-up study since we were better able to control for potential confounds in the lab, and indeed the sample size was larger for the in-lab portion of the study. But we believe that these preliminary results may shed light on how consoling touch is shaping emotional experiences over time. We have been careful to frame the results of the follow-up study as preliminary (page 6, page 21), but believe these results are worth sharing since they align with other research on the delayed benefits of some forms of affect regulation (Kircanski et al., 2012), and thus can inform future studies on this topic of research.

Comment 3: The relationship quality measure and analyses are so lacking in detail that it can’t be interpreted as is. Potentially this can be resolved simply by more details in the manuscript. However, as it is reported in the manuscript now, it is not appropriate for the analyses conducted.

Response 3: Thank you for flagging this area of our manuscript that could use more details. As suggested, we have appropriately renamed the Funk Scale as the Couples Satisfaction Index (CSI), included information about the version and number of items that we used, and provided the full scale in our OSF repository, described on page 13:

“As a measure of relationship satisfaction, participants completed the 32-item version of the Couples Satisfaction Index (CSI), which includes items such as “I have a warm and comfortable relationship with my partner” and “I really feel like part of a team with my partner” [38]. The full scale can be accessed through our OSF repository.”

We have additionally added more information about the models tested on page 15:

“Both of these models (assessing the effect of consoling touch on participants’ emotional pain and comfort) were then re-run with participants’ relationship satisfaction (i.e. CSI) as a possible moderator of the relationship between consoling touch and emotional pain/comfort. These models included valence, touch, relationship satisfaction, and interactions between them as predictors, and emotional pain at first recall as a covariate. We followed up on significant interaction terms that included relationship satisfaction by obtaining estimated marginal means for our model using the “emmeans” package. This method uses the given model to approximate the outcome variable at different levels of a continuous moderator, adjusting for other variables in a model [39].” 

Comment 4: Because of these confounds, this paper is essentially contributing 1) the experimental manipulation of negative memories was more distressing than neutral memories—no condition (handholding) differences, 2) participants felt more comfort from spousal handholding than from control (squeeze-ball) during both negative and emotional memories. Neither of these findings are particularly noteworthy nor do they meaningfully contribute to the literature.

Response 4: Respectfully, we disagree that these findings do not meaningfully contribute to the literature. There is a huge literature on the effects of consoling touch on physical pain, as well as the overlap between physical and emotional pain. However, only a few studies have studied consoling touch in the context of emotional pain (e.g. Kraus, 2019), and no work to date has examined whether consoling touch actually reduces subjective feelings of emotional pain as with physical pain. In everyday life, we often seek and provide emotional support using touch, and emotional pain has tremendous impact on individuals’ wellbeing. We have added more discussion of why it is valuable to examine how touch imparts emotional benefits in the introduction on page 3:

“Three out of four people report that their most painful life experience was emotional in nature, rather than physically painful [1]. Emotional pain, defined as an unpleasant feeling (or suffering) associated with a psychological, non-physical origin, often stemming from thwarted psychological needs [2], undergirds a range of psychiatric issues, including depression, anxiety, borderline personality disorder, and suicidal ideation [3,4]. Given the prevalence of emotional pain, and the negative outcomes associated with such pain, it is crucial to examine how individuals effectively cope with and process it. 

When we experience negative events or hardships, support from others can mitigate the harmful effects of those experiences and buffer us from trauma or prolonged distress [5]. For example, talking through our problems or finding a welcome distraction during a tough time can be valuable in helping us to regulate our emotions [6]. But there are also powerful forms of social support that more implicitly communicate understanding and concern, such as when a loved one holds our hand [7,8]. This type of physical support, often referred to as consoling touch, is observed across species and across cultures [9], and has been shown to reliably reduce the experience of physical pain [10–12]. Notably, however, research has yet to experimentally assess whether touch reduces the subjective experience of emotional pain in the same way that it reduces the subjective experience of physical pain.”

Intuitively, we may think that consoling touch reduces negative affect. On the contrary, however, our findings suggest that touch does not reduce the subjective experience of emotional pain as it does the subjective experience of physical pain. We have underscored this point in our discussion on page 19-20:

“A robust body of work demonstrates that consoling touch can reduce the affective experience of physical pain [10–12], and that physical pain and emotional pain share a common neural system [22–24, c.f. 25]. Intuitively, then, we may assume that consoling touch reduces subjective reports of emotional pain. Surprisingly, however, our results indicate that consoling touch does not decrease the immediate subjective experience of emotional pain relative to holding a squeeze ball in the presence of one’s romantic partner. But, it does lead individuals to feel more comforted by their partner than holding a squeeze ball, particularly when they have greater relationship satisfaction with their partner. This finding is in line with other work suggesting that consoling touch increases subjective feelings of comfort during emotional pain [28], and that relationship satisfaction plays an important role in how we feel during consoling touch [10].” 

One reason this set of findings is noteworthy is that it highlights a possibly important difference between emotional and physical pain. It might not be adaptive to fully attenuate emotional pain, especially the deeply personal pain evoked in this study, because feeling this pain may allow people to process these experiences and more successfully cope with them over time. Meanwhile, it may be more adaptive to down-regulate the experience of physical pain, particularly when people elect to experience that pain in a lab setting. We elaborate on this idea on page 21:

“This set of findings suggests a potentially important difference between emotional pain and physical pain in terms of how negative experiences are regulated in these two contexts. During physical pain, it may be adaptive to down-regulate immediate distress, particularly in a lab setting where the pain is not necessarily helpful in recognizing and escaping some sort of threat. However, during emotional pain, down-regulating immediate distress may not always be adaptive since such distress may be stemming from personally meaningful events that need to be processed and reflected on over time. Indeed, research suggests that individuals often feel motivated to experience negative emotional states because they are helpful in navigating certain experiences (e.g. anger when preparing for a conflict, sadness in coping with a loss), even if those emotional states feel unpleasant [40–42]. Thus, subjective distress may be necessary to some extent to process emotional memories in a way that supports adaptive long-term outcomes and resolution. This idea is consistent with research demonstrating that exposure to negative emotional stimuli (e.g. spiders for phobic patients) can be instrumental in reflecting on and changing harmful cognitions associated with those stimuli over time [43,44]. In other words, we may need to feel certain emotions in order to process and learn from them, allowing us to heal and regulate over time.”

Comment 5: The introduction has some holes as it relates to the research questions investigated. One of the primary stated hypotheses has to do with relationship quality, however, I was greatly wanting for any literature that discusses this topic. The only info listed is a study from nearly 15 years ago... This is not sufficient (other intro holes are addressed in the minor concerns below).

Response 5: We believe it is a fairly intuitive hypothesis that handholding with one’s relationship partner could be influenced by their satisfaction with that partner. Indeed, studies involving romantic couples typically attain some measure of relationship satisfaction/quality (e.g. Reddan, 2020; Conradi et al., 2020). We have clarified this in the introduction, and included additional references to support this hypothesis on page 5-6:

“Since relationship satisfaction often moderates the effect of social support on wellbeing outcomes [29–31], including the effect of handholding on the experience of physical pain [10], we additionally hypothesized that relationship satisfaction would play a moderating role in the effect of touch on emotional pain and comfort, such that greater relationship satisfaction would enhance the soothing effects of touch.”

Comment 6: Your sample size was vastly shrinking by the end of the study. It started out at a decently sized 60 dyads, then down to 47 for the session one and two analyses and finally only 31 dyads for the follow-up survey analyses. That is a pretty small sample size for the follow-up analyses, which is where the contribution of this paper hinges. Because of this, you are no longer examining your primary, controlled experiment data. These follow-up analyses would require more power to detect true differences because of the time elapsed and the self-report recollection nature of the reporting. This is a major confound in your results. Rather than the current rationalization for your sample size (and you can’t really conduct a power analysis to determine your sample size, since that is done), I suggest you run and report an analysis of how likely it was that you observed a significant effect, given your sample, and given your expected effect sizes.

Response 6: We recognize that our sample size is on the smaller side, and this affects our power to detect a true effect, but we hope that you can recognize the complexity of our study design that made it challenging and costly to recruit additional participants. We recruited 60 dyads, a total of 120 participants, for 2 separate in-lab sessions that involved audio-visual recordings, neuroimaging (to collect pilot data for a future study), and running separate but simultaneous protocols for the storyteller and listener. Inevitably, we lost data points from this sample due to technological issues, failure to follow study instructions, and scheduling issues, which we described on page 11:

“Participants were not invited to return to the lab for session 2 if the videos they recorded at session 1 were unusable, either because of technological issues with the recordings or because they did not follow the video prompt instructions. Ten couples were not eligible for session 2 for this reason. Two additional couples dropped out of the study after session 1 due to scheduling issues. One additional couple was removed from analyses due to technological issues during session 2, leaving a total of 47 couples in the sample.”

As per request, post-hoc power analyses were conducted to gain an estimate of whether our sample size was appropriate for the given statistical analyses. We used G*Power for a repeated measures (fully within-subjects) model with 4 conditions (2 x 2 design with valence and touch), a small-medium effect size of 0.25, power of 0.8, and alpha of 0.05. Results indicate that the total sample size recommended is 24 individuals. These analyses suggest that our final sample of 47 dyads for the in-lab study and 31 participants for the follow-up study are reasonable. Of course, future research should aim to replicate these results with a larger sample, and thus we were careful to frame our results as a first step (particularly the follow-up study results). Nonetheless, we believe it is important to share these findings with the scientific community in order to inform future research in this area (a point we elaborated on in Responses 2 and 4). 

We have added some text on page 7 to remind our readers that we used a within-subjects design, and thus are reasonably powered to detect an effect:

“The rationale for our sample size derives from recently published work on affective touch [12,32,33]. Since these studies found effects of touch on pain with samples of 16-43 dyads, we aimed to obtain a sample of 60 dyads for our within-subjects design.”

Comment 7: Continuing on with the follow-up analysis. It could be a pretty big confound to ask people to retrospectively rate how they felt and then at the same time as asking them how they felt now. This could greatly skew your baseline. Why not use the data you already have from them for when they reported how they felt during the experiment as the baseline rather than forcing them to recall how they felt and then imposing that as a baseline?

Response 7: Thank you for this suggestion. As described in Response 2, we re-ran our analyses for this follow-up study using the same baseline as we used in our main analyses as you recommended. We found the same pattern of results when using our baseline measure of emotional pain from the experimental study as we do when using the one item baseline measure from the follow-up study. We have added a description of these analyses on page 19:

“Furthermore, when using our measure of emotional intensity from the in-lab portion of the study (i.e. emotional pain at first recall), b = 0.44, t(57.32) = 3.81, p < .001, 95% CI =[0.21, 0.66], instead of participants’ self-reported emotional pain at the time of the original event assessed at follow-up, we still found a significant effect of touch on current emotional pain, b = -0.59, t(29.76) = -2.15, p = 0.04 95% CI =[-1.12, -0.05].”

Comment 8: A major concern is the way the relationship quality measure is used (potentially just the way you report it?). In the intro, measures, and in the results sections, info is severely lacking. In the measures you only state two items. Was the full scale not used? If not, your relationship quality analyses are greatly confounded; what is the point of having the scale if you only used two items from it? If so, these sections need to be clarified. What are your reliability stats, means, SD, etc. Additionally, how did you use the scale in your analyses? From the figure it looks like you classified couples into high, med, and low RQ… But HOW? How many of your participant-couples fit into each category? Is that same classification used for all your analyses? Additionally, the “Funk measure’s” name is the “Couple Satisfaction Index” or CSI.

Response 8: Thank you for flagging that we should be referring to Funk’s scale as the Couples Satisfaction Index (CSI). As described in Response 3, we now refer to the Funk Scale as the CSI in the paper. We have also included information about the version and number of items that we used, and provided the full scale in our OSF repository, described on page 13. As noted with bolded text below, the items we provide in the paper are simply examples of the scale since the full scale is cited in our references and freely retrievable online (which we now direct our readers to in our OSF repository).

“As a measure of relationship satisfaction, participants completed the 32-item version of the Couples Satisfaction Index (CSI), which includes items such as “I have a warm and comfortable relationship with my partner” and “I really feel like part of a team with my partner” [38]. The full scale can be accessed through our OSF repository.”

We have additionally added more information about the models tested on page 15:

“Both of these models (assessing the effect of consoling touch on participants’ emotional pain and comfort) were then re-run with participants’ relationship satisfaction (i.e. CSI) as a possible moderator of the relationship between consoling touch and emotional pain/comfort. These models included valence, touch, relationship satisfaction, and interactions between them as predictors, and emotional pain at first recall as a covariate.” 

Please note that couples were not classified into high, medium, and low relationship satisfaction categories. Rather, we used estimated marginal means to approximate what participants’ comfort ratings would be at high (mean +1 standard deviation), medium (mean), and low (mean -1 standard deviation) values of relationship satisfaction, after accounting for the other variables in the model. This is a way to break down interactions with a continuous moderating variable without categorically binning dyads into relationship satisfaction groups. We have included reference to the specific package we used (please note that you can review our analyses by viewing our code on OSF), as well as a brief description of this method with citation in our manuscript on page 15:

“We followed up on significant interaction terms that included relationship satisfaction by obtaining estimated marginal means for our model using the “emmeans” package. This method uses the given model to approximate the outcome variable at different levels of a continuous moderator, adjusting for other variables in a model [39].”

Comment 9: Your research paradigm could use some bolstering in how imagining experiences or memories is an appropriate research method. I am not saying it is not appropriate. I am just recommending you address this and justify it with relevant literature. I have listed some suggested references at the bottom that you may want to look into.

Response 9: Thank you for noting this area for clarification. Participants in our study did not simply imagine negative experiences - they were recounting their own past experiences. We did not ask them to imagine what it was like to experience something, but rather to thoroughly describe how a past event from their own life made them feel, and to reflect on how it makes them feel to think about it now. We have expanded on this description on page 8:

“Participants were asked to focus their stories on how they felt at the time of the event, how they dealt with their feelings, and how they feel about the event now. To clarify, participants were not imagining emotional pain. Rather, they were being asked to reflect on and relive their own personal emotionally painful experiences by describing negative events from their past and the feelings those memories brought up in detail [34,35]. This method for manipulating emotion is consistent with countless studies that have used writing or talking about past emotional experiences to evoke an emotional response [36,37], as opposed to using impersonal standardized stimuli to induce negative affect.”

Comment 10: You address how “it is unclear whether touch reduces emotional pain the same as physical pain”. But you never detail why emotional pain matters. You should provide more background and justification on why emotional pain is important and a relevant construct separate from and beyond physical pain. Why should readers care about emotional pain? Addressing this will give the readers a clearer view of why this study matters.

Response 10: Thank you for flagging this portion of the introduction where additional information could be helpful to our readers. We have expanded our discussion of emotional pain and why it matters on page 3:

“Three out of four people report that their most painful life experience was emotional in nature, rather than physically painful [1]. Emotional pain, defined as an unpleasant feeling (or suffering) associated with a psychological, non-physical origin, often stemming from thwarted psychological needs [2], undergirds a range of psychiatric issues, including depression, anxiety, borderline personality disorder, and suicidal ideation [3,4]. Given the prevalence of emotional pain, and the negative outcomes associated with such pain, it is crucial to examine how individuals effectively cope with and process it. 

When we experience negative events or hardships, support from others can mitigate the harmful effects of those experiences and buffer us from trauma or prolonged distress [5]. For example, talking through our problems or finding a welcome distraction during a tough time can be valuable in helping us to regulate our emotions [6]. But there are also powerful forms of social support that more implicitly communicate understanding and concern, such as when a loved one holds our hand [7,8]. This type of physical support, often referred to as consoling touch, is observed across species and across cultures [9], and has been shown to reliably reduce the experience of physical pain [10–12]. Notably, however, research has yet to experimentally assess whether touch reduces the subjective experience of emotional pain in the same way that it reduces the subjective experience of physical pain.”

Comment 11: Switch the order in presentation of method section. Logically before participants tell about their stories, we should be told how they chose which stories to tell.

Response 11: Thank you for this suggestion. We have switched the order of presentation of this information on page 7-8:

“Then, the person assigned to receive support recounted stories about past experiences. For these stories, participants began by completing a form that allowed us to select which negative stories they would recount based on: (a) whether the experience was emotionally painful at the time of the event and (b) whether they were comfortable discussing the experience on camera. This form is available on our OSF repository (titled: “Story Selection”). After selecting which stories they would share, participants recounted a total of 4-5 unrehearsed stories, each lasting about 3 minutes, as we video recorded them.”

Comment 12: How was the prompt on the computer monitor programmed/controlled? Did you use e-prime or some other software that ran automatically? Did you have RA’s running it in the background?

Response 12: Thank you for noting this area for clarification. We have updated our description of Session 1 on page 8 to include more information:

“Reminders for each prompt were presented via Qualtrics. Once the participant was ready to tell their story, they would flip a 3-minute hourglass to help them keep track of time and speak towards the camera. After each story, they responded to questions on Qualtrics about how they felt while sharing the stories.”

We have additionally added information on page 9 about how data was presented at Session 9:

“Prior to each video, they were cued via PsychoPy to either hold hands or hold a squeeze ball… After the rest, they heard a beep that cued them to turn their attention to their laptops to answer questions via Qualtrics about their feelings...”

Comment 13: p.7 you describe how participants fill out a form which RA’s then selected from to determine which story they would tell. More detail on this form should be included. Possibly just including the form as supplementary material.

Response 13: We have added this form under “Study Materials” in our OSF repository, and added a sentence on page 8 to alert the readers of this:

“This form is available on our OSF repository under Study Materials (titled: “Story Selection”).”

Comment 14: p.8, first paragraph: What is meant by “After each story, they responded to questions on the computer monitor about how they felt while sharing the stories.”? Is this part of questions reported in your measures? If so, which measure are you referring to? Or are these questions something different? If so, what does “how they felt” mean or entail? How they felt is too vague.

Response 14: We have added a sentence on page 10 to clarify which questions we are referring to here:

“After each video, participants underwent a minute of “rest” which involved closing their eyes as they continued to hold their partner’s hand or hold the squeeze ball. After the rest, they heard a beep that cued them to turn their attention to their laptops to answer questions via Qualtrics about their feelings, including “how much pain”, “how hurt”, “how sad”, “how angry”, “how much stress or anxiety”, “how emotional”, and “how comforted by their partner” they felt on a Likert scale of 1 to 10. The first six items were used to measure “emotional pain during the task”, whereas the last item measured “comfort during the task” (see Measures for more detail).”

Comment 15: p.8 “two experimenters independently watched the set of videos… and selected two negative videos for viewing in session two”. Were there specific coding by the RA’s to determine if they were emotional? What were the criteria the RA’s used for this decision? Was there any interrater reliability scores you can report? You should have some sort of justification on how their subjective opinion on “emotional” be validated or justified?

Response 15: Thank you for flagging this area of the methods that could use more detail. We have updated the manuscript on page 9 to offer more details about this rating process, and have shared our full protocol for this procedure in our OSF repository (SIC_Pre-Session2_Protocol.pdf). 

“Before session 2, two experimenters independently watched the set of videos to ensure participants appropriately followed the study instructions. As they watched each video, they were asked to provide an overall rating on a scale of 1-10 (10 being the highest) based on the following question: “To what extent did the participant experience emotional pain in the video?” For videos to be considered similar enough for our experimental manipulation in session 2, ratings had to be within at least two points of each other (e.g. 9 and 7). Each rater selected the two videos that they rated as most similar based on the above question. Videos additionally had to be approximately the same length (within 18 s, about 10 seconds of the total video length). If the two raters agreed on which two videos were most similar, those videos were prepared for use in session 2. If the raters disagreed (approximately 22% of the time), a third rater was asked to provide a rating. If no consensus was reached, or no two videos approximately matched on emotionality and length, the couple was excluded from participating in session 2 (see Exclusion Criteria for details).”

Comment 16: Measures section, p. 11 you use the term “’relationship’ between participants’ distress and comfort”. This term should be ‘relation’. Relationship means that thing between real people. Inanimate objects have a ‘relation’ not a ‘relationship’. Especially since you have “relationship quality” as a primary term in your study. That could get confusing.

Response 16: We have rephrased the sentence, now on page 12:

“To test the association between participants’ emotional pain and comfort, we examined the correlation between these variables in each condition.”

Comment 17: I am guessing that your Comfort scale is on the same 1-10 scale as your distress scales was? This should be clarified in that section.

Response 17: We have updated the sentence on page 12:

“Meanwhile, “comfort during the task” was assessed with a single item asking participants how comforted they felt by their romantic partners as they recalled each memory on a scale of 1 to 10.”

Reviewer #2:

Comment 1: This paper, “The Comfort in Touch: Immediate and Lasting Effects of Handholding on Emotional Pain” investigates whether handholding (from a romantic partner) alleviates negative affect associated with the recall of negative life experiences. The authors test for effects in static self-report measures collected both immediately after emotion induction and a few months later. The authors find an effect of handholding on emotional distress in the follow-up period, but not during the task. Handholding did, however, increase feelings of comfort during the task. The subject of this paper interesting, the paradigm novel, and the results push research on the protective effects of touch forward. The study is well powered and gender balanced (which is rare for touch-analgesia research). I do, however, think it is problematic to rely entirely on subjective self-reports especially when it is noted that NIRs data were collected. Self-report measures are easily influenced by experimental demand characteristics and also do not give us much insight into neuropsychological mechanisms. Furthermore, the stimuli are complex time series data but psychological activity during stimulus presentation is entirely disregarded. These interesting and dynamic stimuli are instead reduced to single self-report values post trial. I think the paper requires some major revisions, and I expand upon this in detail below. 

Response 1: We thank you for your feedback and the suggestions for improving upon our manuscript. We agree that neuroimaging data has the potential to inform the mechanisms underlying these phenomena, which is why we collected this pilot neural data, and why we aim to extend this line of research using fMRI. However, please keep in mind that NIRS does not provide access to subcortical neural regions that are informative with regards to the experience of emotional pain, and thus our best measure of emotional pain in this study is from participants’ self-report. Additionally, even though self-report measures are influenced by demand characteristics, we’d like to point out that participants did not rate their pain as lower in the consoling touch condition than the pain control condition (i.e. holding a squeeze ball during a negative video). This finding suggests that participants are not simply telling us what we want to hear in this study, since this result is actually counter to our hypothesis. And while neuroimaging data is important in helping us unpack mechanisms, we do not believe that what participants tell us they are feeling should be wholly disregarded in favor of what their brains may tell us. Self-report and neuroimaging data can simply reveal different parts of the bigger picture, and our study takes a first step towards painting that picture. 

We also appreciate your point regarding the use of a single post-trial measure as opposed to continuous self-report across stimuli presentation. We considered attaining a continuous measure of emotionality across the task, but we find that this measurement approach can be distracting from the task itself, especially given that participants are simultaneously reliving negative personal experiences, and holding their partners hand (depending on the condition). We wanted participants to be able to immerse themselves in this experience that involves both pain from the task and comfort from their partner, instead of having their attention pulled away by our questions. Thus, we opted to only ask for ratings at the end of each trial. 

Comment 2: This paper is conceptualized as a pain paper, but pain is not studied here. This becomes increasingly problematic as the paper progresses. We are set up to think this paradigm will be nearly identical to those in the pain-domain, however, the stimuli here are high-dimensional personal narrative stories 2-3 min long, whereas pain stimuli tend to be ~7-11 sec stimulations, repeated over many trials with controllable intensity. Then when the results are reported and discussed terms like “emotional pain” and “distress” are sometimes used interchangeably and sometimes used to mean different things. I do appreciate that the authors situated this paper within the context of touch-analgesia research, but I think the introduction does not do justice to the way this paper can converse with that research. I think this happens because negative affect is equated with pain, instead of interrogating both phenomena for what they really are in order to understand the complexity of human experience. I am not suggesting that the connections made with pain research be discarded, but instead, be discussed more critically. For example, some analgesia papers do not just study “pain intensity” but also “pain unpleasantness” which can be considered a more “affective” component of pain than the intensity measure. I think this is a noteworthy bridge to your investigation which explores “unpleasantness” so to speak, but not within any context of physical pain.

Response 2: Thank you for noting an area of the paper that is in need of further clarification. We have more clearly laid out our research question regarding emotional pain in the introduction on page 3, including a precise definition of emotional pain:

“Three out of four people report that their most painful life experience was emotional in nature, rather than physically painful [1]. Emotional pain, defined as an unpleasant feeling (or suffering) associated with a psychological, non-physical origin, often stemming from thwarted psychological needs [2], undergirds a range of psychiatric issues, including depression, anxiety, borderline personality disorder, and suicidal ideation [3,4]. Given the prevalence of emotional pain, and the negative outcomes associated with such pain, it is crucial to examine how individuals effectively cope with and process it. 

When we experience negative events or hardships, support from others can mitigate the harmful effects of those experiences and buffer us from trauma or prolonged distress [5]. For example, talking through our problems or finding a welcome distraction during a tough time can be valuable in helping us to regulate our emotions [6]. But there are also powerful forms of social support that more implicitly communicate understanding and concern, such as when a loved one holds our hand [7,8]. This type of physical support, often referred to as consoling touch, is observed across species and across cultures [9], and has been shown to reliably reduce the experience of physical pain [10–12]. Notably, however, research has yet to experimentally assess whether touch reduces the subjective experience of emotional pain in the same way that it reduces the subjective experience of physical pain.”

Additionally, we now include a brief discussion on page 4 of the sensory versus affective components of physical pain (see bolded text), and describe where emotional pain fits into this picture: 

“A body of research suggests that physical pain and emotional pain share a common neural system [22–24], although the extent of this overlap is still debated. Specifically, while physical pain versus no physical pain, and emotional pain (i.e. social rejection) versus no emotional pain are differentiated in the same regions of the brain, physical and emotional pain are represented differently in those regions [25]. Nonetheless, such neural overlap between physical and emotional pain suggests that touch may similarly regulate physical and emotional pain. Additionally, while physical pain and emotional pain differ insofar as physical pain has a sensory component (e.g. stinging, burning) [26] and emotional pain stems from psychological events rather than physical stimulation [2], they both involve an affective component (e.g. unpleasantness, distress). Prior research suggests that consoling touch reduces the affective component of physical pain [27], suggesting that consoling touch also has the potential to reduce the subjective unpleasantness associated with emotional pain.”

Comment 3: Remove the term “emotional pain” from this study entirely. It is not useful to call it “pain” here, it just confuses it with physical pain. I think “distress” is a good term to use throughout. Please operationalize this at the start, and use it consistently (this is particularly confusing in Figures 1 and 3 where one talks about ‘distress’ and the other ‘emotional pain’ but these seems to be the same if not highly correlated measures).

Response 3: We appreciate this reviewer’s perspective, but argue that it is important to keep the term “emotional pain” here. We do not think that readers will confuse emotional pain with physical pain. Indeed, “emotional pain” is a term that is used in common English to refer to suffering or distress related to emotional situations. Moreover, a Google Scholar search of the term “emotional pain” yields over 68,000 entries. Thus, we are not the first to use this term and would like to keep it here in order to connect with other prior literature that has used this same term (e.g., Shneidman, 1996, The Suicidal Mind). However, to more fully clarify how we are using this term, we have now included a definition of this term in our introduction, and more clearly delineated the reasons why we are focusing on emotional pain for this study, shown in Response 2. We also agree that streamlining our language would be helpful for the reader, so we have chosen to use “emotional pain” throughout, instead of more vaguely using the term “pain” or “distress”.

Comment 4: At the end of the introduction revisit this sentence: “Given that physical pain is often temporally bound (i.e. restricted to a certain amount of time), whereas emotional pain is often more enduring, it is possible that physical and emotional pain differ in terms of how consoling touch shapes their immediate and lasting experience.” I think this is a great point, but the way this is written it sounds like it is the *only* difference between them. There are other important differences relevant to this study that can help us to better understand the mechanism supporting touch analgesia. For example, physical pain requires nociceptive input (except in the case of chronic pain). Touch could mediate effects in the periphery or could instead mediate effects at a psychological level. Your study excludes the periphery and therefore could be a nice new piece of evidence to put into conversation with the existing literature.

Response 4: Thank you for noting that there are several interesting distinctions between physical and emotional pain. We have softened our claim in the introduction on page 5 to note that we are simply describing one notable difference that seems relevant to our findings:

“One (though not the only) notable difference between emotional and physical pain is that physical pain is often temporally bound (i.e. restricted to a certain amount of time), whereas emotional pain is often more enduring. Thus, it is possible that physical and emotional pain differ in terms of how consoling touch shapes their immediate and lasting experience.”

Comment 5: On page 4 “A body of research suggests that physical pain and emotional pain share a common neural system (Eisenberger & Lieberman, 2004, 2005; Eisenberger, Lieberman, & Williams, 2003).” The degree to which these two neural processes are the same this is hotly debated, and it is necessary to acknowledge this debate with appropriate citations for the opposing camp (i.e., Wager et al and all the letters involved in the “Pain in the ACC?” debate) if you do want to discuss this work. 

Response 5: Thank you for noting that mention of both sides of this argument would be helpful here. We now speak to this debate on page 4, with additional information about the similarities and differences between physical and emotional pain:

“A body of research suggests that physical pain and emotional pain share a common neural system [22–24], although the extent of this overlap is still debated. Specifically, while physical pain versus no physical pain, and emotional pain (i.e. social rejection) versus no emotional pain are differentiated in the same regions of the brain, physical and emotional pain are represented differently in those regions [25]. Nonetheless, such neural overlap between physical and emotional pain suggests that touch may similarly regulate physical and emotional pain. Additionally, while physical pain and emotional pain differ insofar as physical pain has a sensory component (e.g. stinging, burning) [26] and emotional pain stems from psychological events rather than physical stimulation [2], they both involve an affective component (e.g. unpleasantness, distress). Prior research suggests that consoling touch reduces the affective component of physical pain [27], suggesting that consoling touch also has the potential to reduce the subjective unpleasantness associated with emotional pain.”

Comment 6: Refrain from the term “consoling touch” when describing this investigation — call it what it is — handholding, but link this to the wider concept of consoling touch as appropriate.

Response 6: In order to connect this work with a larger literature on consoling touch across both humans and animals, we think it is important to continue to use the term “consoling touch” to describe the concept that we are examining here. In other words, we think that our study is part of a broader discussion on how different kinds of consoling touch can affect negative emotional experiences. Handholding is simply one way to operationalize consoling touch, as opposed to other forms of touch such as hugging or gentle stroking. We have clarified this point in the introduction on page 5 with the bolded text below:

“This study applied a novel approach to understanding the emotional benefits of touch by examining how handholding with a romantic partner, one form of consoling touch, shapes experiences of emotional pain and comfort during emotional recollection, as well as how it shapes lasting emotional pain associated with emotional experiences.”

Comment 7: “This work suggests that handholding, especially with a romantic partner, attenuates subjective distress associated with physical pain, as well as activation in neural regions associated with threat responses (Coan et al., 2013; Coan et al., 2006; Johnson et al., 2013), with some work suggesting that handholding is more effective than other forms of touch, such as gentle stroking, in reducing subjective pain (Reddan, Young, Falkner, López-Solà, & Wager, 2020).” I would be a little careful with the last claim because that Reddan (2020) paper did not find significant differences between the touch conditions on self-reported pain intensity.

Response 7: Thank you for noting this point. We have removed the latter claim on page 4, and included additional references:

“While research suggests that touch can increase positive feelings like security, and decrease negative feelings like stress [19], the majority of research on the pain-relieving effects of touch have focused on how consoling touch affects individuals experiencing physical pain, such as treatment-related pain, or painful shocks administered in experimental settings. This work suggests that handholding, especially with a romantic partner, attenuates subjective distress associated with physical pain, as well as activation in neural regions associated with threat responses [10–12,20,21].” 

Comment 8: Consider adding evidence from other neuroimaging papers in addition to the Kraus (2019) to give a more complete survey of what we know about handholding analgesia. The Reddan (2020) paper in particular agrees with your hypothesis about safety conceptualization you put forth in the discussion. There is also:

 López-Solà, M., Geuter, S., Koban, L., Coan, J.A., Wager, T.D. (2019). Brain mechanisms of social touch-induced analgesia. Pain. 160(9), 2072–2085.

Response 8: We have added reference to all three of these papers in our introduction (Reference 20, 21, 28). 

Comment 9: Personal narratives have a lot of variation both within a single story and then also across participants. Emotional content and intensity will fluctuate and sometimes people add a “silver lining” or what they learned from a trying event at the end of their recollection which then can make the story somewhat “positive.” In short, these stimuli are complex! But this complexity is not controlled for or even assessed in any meaningful way. I appreciate that the authors attempted to control for “emotional intensity” in their stimuli via a “memory-related distress” score collected during visit 1, however, I am not convinced this control is valid.

Response 9: We appreciate the points you are making, but a natural consequence of using naturalistic stimuli is that they will be more variable and complex than standardized impersonal stimuli. We wanted to use personal stimuli for this study to more closely approximate what it is like to receive emotional support in the form of touch in everyday life. We often share our personal experiences with close others, and this may create opportunities for them to offer support through consoling touch. We agree that emotional content and intensity fluctuate between individuals, and between emotional events, but this is why we used a within-subjects design. Since our measure of emotional pain at first recall (previously called “memory-related distress”) captures how painful each story was for participants without a touch manipulation, we believe this score is the best way to control for variation in the emotional intensity of each video (from participants’ own perspective).

We would also like to point out that we had independent raters view and rate the stories to ensure they were not too dissimilar in terms of perceived emotional intensity before inviting participants to return for participation in session 2. We have updated the manuscript on page 9 to offer more details about this rating process, and have shared our full protocol for this procedure in our OSF repository (SIC_Pre-Session2_Protocol.pdf). 

“Before session 2, two experimenters independently watched the set of videos to ensure participants appropriately followed the study instructions. As they watched each video, they were asked to provide an overall rating on a scale of 1-10 (10 being the highest) based on the following question: “To what extent did the participant experience emotional pain in the video?” For videos to be considered similar enough for our experimental manipulation in session 2, ratings had to be within at least two points of each other (e.g. 9 and 7). Each rater selected the two videos that they rated as most similar based on the above question. Videos additionally had to be approximately the same length (within 18 s, about 10 seconds of the total video length). If the two raters agreed on which two videos were most similar, those videos were prepared for use in session 2. If the raters disagreed (approximately 22% of the time), a third rater was asked to provide a rating. If no consensus was reached, or no two videos approximately matched on emotionality and length, the couple was excluded from participating in session 2 (see Exclusion Criteria for details).”

Comment 10: Being that the “memory-related distress” score and the “task-related distress” score are the same questions asked of the same people about the same stimuli, I expect these to be highly correlated. Are they? Please test for multicollinearity in all the models which you use the “memory-related distress” score as a nuisance regressor (or the measure where they “recalled how they felt at the time of the event”). If it is correlated with your regressor of interest then it is possible your reported tests are invalid (and this might not be a bad thing — maybe your null findings are actually detectable). 

Response 10: Thank you for this feedback. Yes, memory-related distress (now referred to as “emotional pain at first recall”) and task-based distress (now referred to as “emotional pain during the task”) were correlated (r = 0.73 in the consoling touch condition and r = 0.55 in the emotional pain only condition). This makes sense since baseline covariates tend to be correlated with outcome variables (e.g. pre and post-intervention math scores should be correlated). However, multicollinearity is an issue that characterizes redundancy in predictor variables (James et al. 2014). Task-related distress (i.e. emotional pain during the task) is our outcome variable, not a predictor variable. Thus, correlation between task-related distress and memory-related distress does not cause an issue of multicollinearity. For thoroughness, we went ahead and tested for multicollinearity in all of our models (using the “car” package in R), and found no evidence of multicollinearity (all VIF scores were < 4). 

Comment 11: Can you attempt to provide some kind of validity check across the stimuli alone? You can, for example, get independent raters to rate the videos moment-by-moment. They you can see if these stimuli are balanced across participants. You can look for the amount of negativity vs like “silver lining” in the negative stories and test if there are outlier stories, etc. This might require you collecting more data on these videos, and I know that can be a big and annoying ask, but I think it is important and won’t be wasted. You can use those “normative” ratings in future studies, etc. Also, if you know these stimuli are balanced you can eliminate the use of that “memory-related distress” nuisance regressor which might be watering down your effects.

Response 11: Thank you for flagging this area of the methods that could use clarification. As described in Response 9, we had our research team rate these videos on overall emotional intensity to help with balancing the conditions. As you brought up in Comment 9, it is very difficult to perfectly match stimuli across conditions when using personal stimuli, but we have argued that there are benefits to using personalized stimuli that justify our use of them in this study. We know our stimuli are not perfectly matched, but this is why we controlled for memory-related distress (i.e. emotional pain at first recall) in our analyses. We believe this measure was the best way to control for variation in the emotional intensity of each video, from participants’ own perspective. 

Comment 12: Was the follow up survey a post hoc addition or part of the original study design? Please specify both at the time when it is first introduced and when it is described in the methods. Address the length of time/gap as well, earlier on. 

Response 12: We have clarified this point on page 6 and 11:

“Finally, participants completed an exploratory follow-up survey to assess whether there were any lasting effects of handholding on the experience of emotional pain. In other words, we aimed to test whether emotional memories paired with handholding in the lab would later be recalled with less emotional pain than those that were paired with holding a squeeze-ball. Given the exploratory nature of this follow-up study, this data is considered preliminary in elucidating how consoling touch potentially shapes the lasting experience of emotional pain.”

“To examine potential lasting effects of consoling touch on emotional pain, participants who received support completed a brief exploratory follow-up survey. These surveys were sent out electronically after we completed in-lab data collection for all of our dyads as an exploratory addition to our investigation. Thus, participants completed the survey between 1.28 and 7.82 months after session 2 (M = 4.01 months). ”

Comment 13: Can you clarify whether the supportive partner could hear or see the story/stimulus beyond the curtain and importantly whether the support receiver was aware of this (could the support receiver possibly think “We are watching this/experiencing this together?”)

Response 13: Thank you for noting this area for clarification. Yes, the support giver could hear and see the stimuli, as the stimuli presentation screen was situated in the center of the room across from both participants. Thus, the support receiver was aware that they were experiencing the stimuli with their partner. However, the stimuli was specifically personal to the support receiver (e.g. a time they felt rejected or alone), so the support giver would have had a very different experience of the stimuli than the support receiver. Participants knew which one of them was the “storyteller” and the “listener”, so there would have been no confusion about who was in a position to provide support. We have added some information about this on page 10 (see bolded text):

“Throughout the task, participants sat on opposite sides of a curtain from each other to prevent them from communicating verbally or through other non-verbal cues (e.g. body language, facial expressions). Both participants could hear and see the videos on a single screen on the wall across from them, such that the support receiver and support giver experienced the stimuli simultaneously. During the two hand-holding conditions, they held hands through the curtain. Thus, participants were aware of the presence of their partner in all four conditions, but their physical contact was limited to the two conditions that included touch.”

Comment 14: “The final sample included predominantly Caucasian (39%) and Asian/Asian American (34%) participants (mean age = 21.75 years). “ Please give the complete racial breakdown.

Response 14: We have added detailed reporting of the ethnicity breakdown in the manuscript on page 6. Please note that these percentages changed a bit when creating a separate category for multiracial participants. 

“The final sample included approximately 30% White, 32% Asian/Asian American, 11% Latino/a, 2% Filipino/a, 2% Black, and 11% multiracial participants. The remaining participants chose another identity or preferred not to answer.”

Comment 15: Can you clarify if you measured gender identity or biological sex?

Response 15: We measured gender identity through the following question: “What is your gender?” Thus, we are careful to use the term gender rather than sex throughout the manuscript. 

Comment 16: make clear if the scales were Likert or VAS (It seems like Likert).

Response 16: We have specified the use of Likert scales on page 10:

“...including “how much pain”, “how hurt”, “how sad”, “how angry”, “how much stress or anxiety”, “how emotional”, and “how comforted by their partner” they felt on a Likert scale of 1 to 10.” 

Comment 17: The term “memory-related distress” is confusing and not ideal.

Response 17: We agree and have replaced this term throughout with “emotional pain at first recall.” Please see our note (“Note to Editor and all Reviewers”) in the beginning of this document for details.

Comment 18: For clarity, please use a different shorthand for your conditions - not “touch-pain condition” etc., to indicate this paradigm is about emotional distress and not physical pain.

Response 18: We agree and have replaced these condition names throughout to minimize confusion (see below text from page 9-10 for example):

“Prior to each video, they were cued via PsychoPy to either hold hands or hold a squeeze ball such that participants underwent four conditions in a randomized order: (a) hand-holding during a negative video (i.e. consoling touch condition); (b) hand-holding during a neutral video (i.e. touch only control condition); (c) holding a squeeze ball during a negative video (emotional pain only control condition); and (d) holding a squeeze ball during a neutral video (full control condition).”

Comment 19: Please include a results table for the non-significant GLMM gender test in your supplementary

Response 19: We have added these results to our Supplementary Materials and referenced these materials on page 14.

Comment 20: Please show the data distribution/individual data points in you bar plots

Response 20: We have revised figures 1 and 3 to use box plots instead of bar plots to give readers a better sense of the distribution. See below (Figures 1A, 1B, and 3):

Comment 21: Report complete stats for the following two post hoc tests that only report p-values “when additionally controlling for the amount of time that passed between the in-lab manipulation and the completion of the follow-up survey, this effect of touch on current emotional pain did not change (p = 0.03). Relationship satisfaction was not a significant moderator of this effect (p = 0.67).”

Response 21: We have expanded the description of these results on page 18-19:

“When including relationship satisfaction as a moderator in the model, we did not find a significant interaction between touch and relationship satisfaction, b = -0.02, t(28.06) = -0.75, p = 0.46, 95% CI = [-0.06, 0.02].

In addition, when controlling for emotional pain at the time of the original emotional event, b = 0.50, t(53.38) = 5.33, p < .001, 95% CI =[0.32, 0.69], and the amount of time that passed between the in-lab manipulation and the completion of the follow-up survey, b = -0.00, t(28.35) = -0.78, p = 0.44, 95% CI =[-0.01, 0.00], we still found a significant effect of touch on current emotional pain, b = -0.58, t(29.05) = -2.24, p = 0.03, 95% CI =[-1.09, -0.07]. Furthermore, when using our measure of emotional intensity from the in-lab portion of the study (i.e. emotional pain at first recall), b = 0.44, t(57.32) = 3.81, p < .001, 95% CI =[0.21, 0.66], instead of participants’ self-reported emotional pain at the time of the original event assessed at follow-up, we still found a significant effect of touch on current emotional pain, b = -0.59, t(29.76) = -2.15, p = 0.04 95% CI =[-1.12, -0.05].”

 

Comment 22: There is a history of gender effects in the touch-pain studies you mention (i.e., Coan and Lopez-Sola studies use ONLY female samples, Reddan found gender effects, etc). You found no effects of gender. Please consider discussing this (can be brief).

Response 22: We have added a brief description about this point on page 20 of the discussion:

“We found no effect of gender on our outcome variables. Given that the majority of touch studies have only examined female participants [10,12,21], with some work finding gender differences during the experience of physical pain [20], further research is needed to clarify how gender shapes the outcome of consoling touch.”

Comment 23: Please reword the following sentence, it is confusing: “Indeed, research suggests that individuals are often motivated to experience certain emotional states even if they feel hedonically unpleasant, because they are in some way instrumental to feel.” 

Response 23: Thank you for this note. We have edited this sentence on page 21:

“Indeed, research suggests that individuals often feel motivated to experience negative emotional states because they are helpful in navigating certain experiences (e.g. anger when preparing for a conflict, sadness in coping with a loss), even if those emotional states feel unpleasant [40–42].”

Comment 24: Please reword “In other words, we may need to feel certain emotions in order to process and overcome them. “Maybe make this sentence more concrete. I am not sure emotions are something to be “overcome” but maybe bad beliefs or persistent memories underlying the emotions are?

Response 24: Thank you for this note. We have edited this sentence on page 21:

“In other words, we may need to feel certain emotions in order to process and learn from them, allowing us to heal and regulate over time.”

Comment 25: Does your IRB require that you restrict data sharing (as described on pg 5 of the compiled manuscript in the "where they data may be found" section)? If not, consider changing this so you can share the data for this or future projects.

Response 25: Our IRB requires that we restrict data sharing as we have, but we have already deposited our data (which are time-stamped) on OSF so that they can be easily shared upon request with other researchers using a private link: https://osf.io/9wbua/?view_only=de32be8c75a94f1e9d3bdb9c7c21f19f

Comment 26: Please considering including the data that was used to make the composite scores in the uploaded/shareable data.

Response 26: We have updated the datafile on our OSF repository to include the individual items used to create composite scores.

Comment 27: When you used Google Hangouts to monitor your participants, were you able to do this in a way that prevented data-sharing with Google? I haven’t seen Hangouts used like this before, and this is just one of those things that makes me think about how our research practices have to evolve alongside tech/data-mining companies. 

Response 27: We did not take any steps to prevent data-sharing with Google, as I’m not sure there is a way to do this while using Google’s services, or that there is a similar free service that would escape this issue. Our participants were fully aware of us using Google Hangouts to monitor them, and agreed to this use of Google Hangouts during the study session. It’s a great point that you make though regarding how our research practices going forward will have to attend to the way that these companies mine data, and we thank you for noting this. 

Comment 28: The mention of the NIRs data being saved for a different study creates concern about dual publication.

Response 28: Thank you for raising this concern. Our NIRs data were used for exploratory analyses in preparation for a future fMRI study, and we do not have plans to publish this pilot data. To clarify this, we have rephrased the sentence on page 6: 

“These neural data were designed to serve as exploratory pilot data for a future neuroimaging study.”

Reviewer #3:

Comment 1: The manuscript is well-written and provides novel insights on the effect of touch on distress and comfort and lasting pain during emotional recollection. I have several issues that should be addressed.

Response 1: We thank you for your feedback and the suggestions for improving upon our manuscript. We have taken steps to address your concerns, detailed below.

Comment 2: Generally, the introduction mostly sounds reasonable. However, the manuscript deals with memory-related pain/distress but this issue was not connected to the proposed rational.

Response 2: Thank you for noting this area for clarification. Participants in our study were asked to thoroughly describe how a past event from their own life made them feel, and to reflect on how it makes them feel to think about it now. Thus, our aim in using their own personal memories was to manipulate their current emotional state. We have expanded on this description on page 7 to more clearly describe why we used their memories to induce emotional pain:

“Participants were asked to focus their stories on how they felt at the time of the event, how they dealt with their feelings, and how they feel about the event now. To clarify, participants were not imagining emotional pain. Rather, they were being asked to reflect on and relive their own personal emotionally painful experiences by describing negative events from their past and the feelings those memories brought up in detail [34,35]. This method for manipulating emotion is consistent with countless studies that have used writing or talking about past emotional experiences to evoke an emotional response [36,37], as opposed to using impersonal standardized stimuli to induce negative affect.”

We have also added additional motivation for our investigation of emotional pain and connected it with the existing literature on page 3:

“Three out of four people report that their most painful life experience was emotional in nature, rather than physically painful [1]. Emotional pain, defined as an unpleasant feeling (or suffering) associated with a psychological, non-physical origin, often stemming from thwarted psychological needs [2], undergirds a range of psychiatric issues, including depression, anxiety, borderline personality disorder, and suicidal ideation [3,4]. Given the prevalence of emotional pain, and the negative outcomes associated with such pain, it is crucial to examine how individuals effectively cope with and process it. 

When we experience negative events or hardships, support from others can mitigate the harmful effects of those experiences and buffer us from trauma or prolonged distress [5]. For example, talking through our problems or finding a welcome distraction during a tough time can be valuable in helping us to regulate our emotions [6]. But there are also powerful forms of social support that more implicitly communicate understanding and concern, such as when a loved one holds our hand [7,8]. This type of physical support, often referred to as consoling touch, is observed across species and across cultures [9], and has been shown to reliably reduce the experience of physical pain [10–12]. Notably, however, research has yet to experimentally assess whether touch reduces the subjective experience of emotional pain in the same way that it reduces the subjective experience of physical pain.”

Comment 3: The description of the analytical approach is very general and not specific to the current study. For example, what residual distribution was assumed? What degrees of freedom were used? What random effects/covariance structures were assumed?

Response 3: We have added more information about our models on page 14:

“For our analyses, we used the statistical package R (Version 1.2.1335) to create linear mixed models (LMMs, i.e. multilevel regression) with participant ID as the group level variable, fixed effects, and random intercepts. We used the “lmer” package in R, which by default uses the Satterthwaite degrees of freedom method and bases confidence intervals and p-values on the t-distribution.”

Comment 4: Please, provide a detailed description of the analysis regarding the gender effect.

Response 4: We have added a full report of these analyses and results to our Supplementary Materials and referenced these materials on page 14 of the manuscript.

---

## [Decision Letter · Decision Letter 1]

20 Jan 2021

PONE-D-20-25552R1

The Comfort in Touch: Immediate and Lasting Effects of Handholding on Emotional Pain

PLOS ONE

Dear Dr. Sahi,

Thank you for submitting your manuscript to PLOS ONE. After careful consideration, we feel that it has merit but does not fully meet PLOS ONE’s publication criteria as it currently stands. Therefore, we invite you to submit a revised version of the manuscript that addresses the points raised during the review process.

Thank you for addressing the reviewer's suggestions and revising your manuscript thoroughly. I appreciate your work and would only draw your attention to some minor suggestions by reviewer 2, which could be addressed, I think.

We look forward to receiving your revised manuscript.

Kind regards,

Hedwig Eisenbarth

Academic Editor

PLOS ONE

Additional Editor Comments (if provided):

Thank you for addressing the reviewer's suggestions and revising your manuscript thoroughly. I appreciate your work and would only draw your attention to some minor suggestions by reviewers 1 and 2, which could be addressed, I think.

Reviewers' comments:

Reviewer's Responses to Questions

**Comments to the Author**

1. If the authors have adequately addressed your comments raised in a previous round of review and you feel that this manuscript is now acceptable for publication, you may indicate that here to bypass the “Comments to the Author” section, enter your conflict of interest statement in the “Confidential to Editor” section, and submit your "Accept" recommendation.

Reviewer #1: All comments have been addressed

Reviewer #2: (No Response)

Reviewer #3: All comments have been addressed

2. Is the manuscript technically sound, and do the data support the conclusions?

Reviewer #1: Yes

Reviewer #2: Partly

Reviewer #3: Yes

3. Has the statistical analysis been performed appropriately and rigorously? 

Reviewer #1: Yes

Reviewer #2: No

Reviewer #3: Yes

4. Have the authors made all data underlying the findings in their manuscript fully available?

Reviewer #1: Yes

Reviewer #2: Yes

Reviewer #3: Yes

5. Is the manuscript presented in an intelligible fashion and written in standard English?

Reviewer #1: Yes

Reviewer #2: Yes

Reviewer #3: Yes

6. Review Comments to the Author

Reviewer #1: The authors were appropriately responsive to my comments and have addressed my concerns within the manuscript. I think this will make a nice addition to the literature.

The only lingering comment I would like to make is regarding the word choice of "relationship" between variables of interest. The one change you already made in the in the revised paper reads cleaner and I would suggest updating the other areas of occurrence. Although this is a very minor comment, I think it helps with clarity. Especially as you have a primary variable of interest as "relationship satisfaction" and these terms are used frequently together. See examples of problematic areas below.

p. 18 "... or the relationship between valence and comfort, b =

0.00, t(128.23) = 0.16, p = 0.88, 95% CI = [-0.04, 0.05], but did significantly moderate

the relationship between touch and comfort, b = 0.07, t(128.27) = 2.96, p = .004, 95% CI

= [0.02, 0.11]."

Additionally p. 15 "... Both of these models (assessing the effect of consoling touch on participants’

emotional pain and comfort) were then re-run with participants’ relationship satisfaction

(i.e. CSI) as a possible moderator of the relationship between consoling touch and

emotional pain/comfort. These models included valence, touch, relationship satisfaction,

and interactions between them as predictors, and emotional pain at first recall as a

covariate."

Reviewer #2: Thank you for your response and revisions. The authors did not fully address some of my original comments that I think are important for the revision. Below they are listed in order of importance.

1 - In your response to comment 12, you did not clarify whether the follow up survey was a post hoc addition or part of the original design. It is critical for the reader to know whether this test was planned a priori or whether it was added on after no effects were found in the main study. It is unclear to me what is meant by exploratory in this context, and this needs to be clarified as this is the finding the entire paper hinges on. That being said, can you redo the analysis/figures from Figures 1A & 1B (During Task) with just the participants that are included in 3 (Emotional Pain at Follow-up) and include this figure and stats in the supplementary? It would be useful to know if there is an effect for hand v ball "during task" in the reduced sample size which shows an effect "at follow-up."

2 - Response to comment 2: I like the authors' new addition to the discussion on pg 4 (“While physical pain and emotional pain differ….” etc.), however, I think the preceding discussion of the common neural systems (“A body of research suggests that physical pain and emotional pain share a common neural system [22–24], although the extent of this overlap is still debated. …. Additionally,”) should be removed or revised again for the following reasons:

(a) The authors did not add any neural or physiological data to their revision, so bringing up this neuroimaging debate is out of place.

(b) The affective components of physical pain are not equivalent to what is being described as “emotional pain” here, so the beginning of this paragraph makes the new addition (which is more pertinent) confusing.

(c) there is an effect of handholding (v squeeze ball) on self-reports of physical pain “during task” that has been replicated-- there is no effect “during task” here. This is not addressed.

(d) Finally, claims like “share a common neural system” are somewhat meaningless since that common neural system is the brain, and “overlap” does not provide evidence of shared regulatory processes.

3 - To comment 1, the authors responded, “However, please keep in mind that NIRS does not provide access to subcortical neural regions that are informative with regards to the experience of emotional pain, and thus our best measure of emotional pain in this study is from participants’ self-report.”

What was the motivation to collected NIRS data if you did not expect to acquire signal relevant to your study? Would you not expect relevant signal in the PFC, considering its role in emotion regulation? In general, I think my point with regards to an over reliance on self-report data was missed. I was not suggesting that self-report is useless - it is not, it captures one's subject experience. This paper, however, is very limited in its contribution to the literature because it relies entirely on subjective self-report.

Reviewer #3: I'm fine with the current version of the manuscript. Merry Christmas, Happy Hanuka and Happy New Year!

7. PLOS authors have the option to publish the peer review history of their article (what does this mean?). If published, this will include your full peer review and any attached files.

Reviewer #1: No

Reviewer #2: No

Reviewer #3: No

---

## [Author Response · Author response to Decision Letter 1]

22 Jan 2021

Round 2 Response to Editor and Reviewers (PONE-D-20-25552R1)

Editor Comments

Comment 1: Thank you for addressing the reviewer's suggestions and revising your manuscript thoroughly. I appreciate your work and would only draw your attention to some minor suggestions by reviewers 1 and 2, which could be addressed, I think.

Response 1: We thank you and the reviewers for your second round of feedback to improve upon our manuscript. We have responded to the remaining minor comments in what follows. We hope you’ll agree that this revised manuscript is now ready to share with the scientific community. 

Reviewer #1:

Comment 1: The authors were appropriately responsive to my comments and have addressed my concerns within the manuscript. I think this will make a nice addition to the literature.

The only lingering comment I would like to make is regarding the word choice of "relationship" between variables of interest. The one change you already made in the in the revised paper reads cleaner and I would suggest updating the other areas of occurrence. Although this is a very minor comment, I think it helps with clarity. Especially as you have a primary variable of interest as "relationship satisfaction" and these terms are used frequently together. See examples of problematic areas below.

p. 18 "... or the relationship between valence and comfort, b =

0.00, t(128.23) = 0.16, p = 0.88, 95% CI = [-0.04, 0.05], but did significantly moderate

the relationship between touch and comfort, b = 0.07, t(128.27) = 2.96, p = .004, 95% CI

= [0.02, 0.11]."

Additionally p. 15 "... Both of these models (assessing the effect of consoling touch on participants’

emotional pain and comfort) were then re-run with participants’ relationship satisfaction

(i.e. CSI) as a possible moderator of the relationship between consoling touch and

emotional pain/comfort. These models included valence, touch, relationship satisfaction,

and interactions between them as predictors, and emotional pain at first recall as a

covariate."

Response 1: We thank you for your feedback overall in improving our manuscript, and for this final note to improve the clarity of our results section. We agree that this change helps these sections read cleaner. The recommended changes have been made:

p. 15: “Both of these models (assessing the effect of consoling touch on participants’ emotional pain and comfort) were then re-run with participants’ relationship satisfaction (i.e. CSI) as a possible moderator of the association between consoling touch and emotional pain/comfort.”

p. 18: “When including participants’ relationship satisfaction scores in the model as a potential moderator, we found that relationship satisfaction did not moderate the association between valence by touch and comfort, b = -0.05, t(128.22) = -1.42, p = 0.16, 95% CI = [-0.11, 0.02], or the association between valence and comfort, b = 0.00, t(128.23) = 0.16, p = 0.88, 95% CI = [-0.04, 0.05], but did significantly moderate the association between touch and comfort, b = 0.07, t(128.27) = 2.96, p = .004, 95% CI = [0.02, 0.11].”

Reviewer #2:

Comment 1: Thank you for your response and revisions. The authors did not fully address some of my original comments that I think are important for the revision. Below they are listed in order of importance. In your response to comment 12, you did not clarify whether the follow up survey was a post hoc addition or part of the original design. It is critical for the reader to know whether this test was planned a priori or whether it was added on after no effects were found in the main study. It is unclear to me what is meant by exploratory in this context, and this needs to be clarified as this is the finding the entire paper hinges on.

Response 1: Thank you for your suggestions to further improve upon our manuscript. We appreciate your point about transparency and have made the nature of this follow-up study more clear on p. 11:

“These surveys were sent out electronically after we completed in-lab data collection for all of our dyads. Thus, participants completed the survey between 1.28 and 7.82 months after session 2 (M = 4.01 months). Because we decided to add this follow-up assessment to our investigation while data collection was ongoing, this portion of the project was an exploratory addition to the original research plan.” 

Comment 2: That being said, can you redo the analysis/figures from Figures 1A & 1B (During Task) with just the participants that are included in 3 (Emotional Pain at Follow-up) and include this figure and stats in the supplementary? It would be useful to know if there is an effect for hand v ball "during task" in the reduced sample size which shows an effect "at follow-up."

Response 2: We appreciate your point about re-running our analyses with the smaller subset of participants from the follow-up study to ensure consistency with our results using the full sample. We have included detailed information about this in the supplement, and referenced it in the manuscript on p. 16:

“Finally, since this follow-up study involved a smaller subset of participants than our primary analyses, we re-ran our primary analyses using this smaller subset of participants to ensure consistency in our results (see Supplementary Materials for a full report of these analyses and accompanying figures).”

Our results from the primary analyses using the full dataset (N = 47) and the reduced dataset (N = 31) are consistent (see p. 2-3 of Supplement):

 “Since our follow-up study was completed by a smaller subset of participants (N = 31) than the full sample used in our primary analyses (N = 47), we re-ran our primary analyses using this smaller subset of participants to ensure consistency in our results. Consistent with our results on the effects of consoling touch on participants’ emotional pain using the full sample, there was a significant main effect of valence, b = -0.81, t(106.55) = -2.69, p = 0.008, 95% CI = [-1.40, -0.23], no main effect of touch, b = -0.19, t(88.58) = -0.85, p = 0.40, 95% CI =[-0.61, 0.24], and no interaction between valence and touch, b = 0.23, t(88.03) = 0.74, p = 0.46, 95% CI =[-0.37, 0.83], on how much emotional pain participants felt, controlling for potential differences in the emotional intensity of the different memories being recalled (i.e. emotional pain at first recall), b = 0.61, t(116.90) = 9.66, p < .001, 95% CI = [0.49, 0.73]. Participants felt significantly more emotional pain during the negative videos (M = 4.01, SD = 1.75) than the neutral videos (M = 1.19, SD = 0.26) (Supplementary Figure 1A). Contrary to our hypothesis, pairwise comparisons indicated no significant difference between how much emotional pain participants felt during the consoling touch condition versus the emotional pain only condition, t(87.9) = 0.85, p = 0.83, 95% CI = [-0.39, 0.77]. 

Additionally, consistent with our results on the effects of consoling touch on participants’ comfort using the full sample, there was no main effect of valence, b = 0.05, t(101.56) = 0.08, p = 0.94, 95% CI = [-1.28, 1.40], a significant main effect of touch, b = 3.12, t(87.16) = 6.38, p < .001, 95% CI = [2.17, 4.06], and no interaction between valence and touch, b = -0.62, t(86.71) = -0.90, p = 0.37, 95% CI = [-1.96, 0.72], on how comforted participants felt by their partner, controlling for potential differences in the emotional intensity of the different memories being recalled (i.e. emotional pain at first recall), b = 0.22, t(112.97) = 1.51, p = 0.13, 95% CI = [-0.07, 0.51]. Participants felt more comforted by holding their partners’ hand (M = 5.45, SD = 2.65) than by holding a squeeze ball (M = 2.60, SD = 2.02) (Supplementary Fig 1B). Consistent with our hypothesis, pairwise comparisons indicated that participants felt significantly more comforted during the consoling touch condition M = 6.13, SD = 2.21) than the emotional pain only condition (M = 2.94, SD = 1.84), t(87.7) = -6.38, p < .001, 95% CI = [-4.40, -1.84].”

Comment 3: I like the authors' new addition to the discussion on pg 4 (“While physical pain and emotional pain differ….” etc.), however, I think the preceding discussion of the common neural systems (“A body of research suggests that physical pain and emotional pain share a common neural system [22–24], although the extent of this overlap is still debated. …. Additionally,”) should be removed or revised again for the following reasons:

(a) The authors did not add any neural or physiological data to their revision, so bringing up this neuroimaging debate is out of place.

(b) The affective components of physical pain are not equivalent to what is being described as “emotional pain” here, so the beginning of this paragraph makes the new addition (which is more pertinent) confusing.

(c) there is an effect of handholding (v squeeze ball) on self-reports of physical pain “during task” that has been replicated-- there is no effect “during task” here. This is not addressed.

(d) Finally, claims like “share a common neural system” are somewhat meaningless since that common neural system is the brain, and “overlap” does not provide evidence of shared regulatory processes.

Response 3: We appreciate your suggestions to streamline our introduction for relavance and have removed the majority of the discussion of the debate on the neural underpinnings of physical versus emotional pain on p. 4 as you now recommend. However, we have modified and retained one sentence to reference the literature on shared neural regions, as well as a reference to the debate on this topic, because we do think it is important to point out that there has been considerable research on this topic, and this neuroimaging research has served as the inspiration and rationale for the current research project. Just as data from animal research can inspire research in humans, and physiological data can inspire a reaction time study, neuroimaging research can inspire a behavioral study. Thus, this neuroimaging research is relavant to our present research question and we believe it should be mentioned in the introduction. Please see the changes made below on p. 4:

“A body of research suggests that physical pain and emotional pain are processed in overlapping neural regions [22–24], although the extent of this overlap is still debated [25]. Specifically, while physical pain versus no physical pain, and emotional pain (i.e. social rejection) versus no emotional pain are differentiated in the same regions of the brain, physical and emotional pain are represented differently in those regions [25]. Nonetheless, such neural overlap between physical and emotional pain suggests that touch may similarly regulate physical and emotional pain. Additionally, While physical pain and emotional pain differ insofar as physical pain has a sensory component (e.g. stinging, burning) [26] and emotional pain stems from psychological events rather than physical stimulation [2], they both involve an affective component (e.g. unpleasantness, distress). Prior research suggests that consoling touch reduces the affective component of physical pain [27], suggesting that consoling touch also has the potential to reduce the subjective unpleasantness associated with emotional pain.”

With regard to point C, that there is no effect of “task” in this study like in studies of physical pain, that is not quite accurate. Studies that examine the effect of handholding on physical pain have two conditions: pain during handholding and pain without handholding. These studies show that self-reported pain is reduced during handholding (vs. no handholding). In the current study, we find that while subjects do not report less emotional pain during the consoling touch condition (emotional pain with touch) as compared to the emotional pain only condition (emotional pain without touch), participants did report higher levels of comfort during the consoling touch condition. Thus, there is an effect of task in the current study, but with a different descriptor (participants express it in terms of more comfort rather than less emotional pain).

Finally, with respect to point D, that claims about “shared neural systems” are meaningless, we respectfully disagree. For example, showing that seeing a stimulus and imagining a stimulus both activate the same region of occipital cortex suggests common processes underlying these two experiences (Kosslyn et al., 1995, Nature). Thus, while this reviewer may not agree that physical and social pain share similar underlying neural circuitry, and this debate is important to reference as we have, we don’t think the idea of shared neural circuitry is meaningless, uninteresting, or unimportant. (Please note that we also changed the language “shared neural system” to “overlapping neural regions” to address any readers who might interpret “shared neural system” as referring to the entire brain; this was not our intended meaning.) 

Comment 4: To comment 1, the authors responded, “However, please keep in mind that NIRS does not provide access to subcortical neural regions that are informative with regards to the experience of emotional pain, and thus our best measure of emotional pain in this study is from participants’ self-report.”

What was the motivation to collected NIRS data if you did not expect to acquire signal relevant to your study? Would you not expect relevant signal in the PFC, considering its role in emotion regulation? In general, I think my point with regards to an over reliance on self-report data was missed. I was not suggesting that self-report is useless - it is not, it captures one's subject experience. This paper, however, is very limited in its contribution to the literature because it relies entirely on subjective self-report.

Response 4: We would like to share more information with you about the NIRS portion of the research to help clarify what we aimed to study, and why we are not sharing this data. We collected NIRS data to begin a preliminary exploration of the neural processes underlying consoling touch during emotional pain to help inform a future fMRI study that we are currently in the process of planning. Specifically, we hyperscanned both participants in each dyad (the person receiving support and the person providing support) to assess whether participants were more “in sync” during consoling touch, similar to Dr. Shamay-Tsoory’s work on inter-partner coupling during physical pain (Goldstein, Weissman-Fogel, & Shamay-Tsoory, 2017). However, we had several issues with data collection that preclude us from reliabily inferring neural activity during the task. Most notably, participant head motion lead the fNIRS caps to slide around on their heads such that the position of the optodes shifted signficantly across the task. Given that we were hyperscanning participants, if even one participant from each dyad had this issue, we were unable to use their data to assess neural synchrony as this motion makes it really difficult to attain spatial specificity in our analyses. As mentioned, we plan to follow-up on this behavioral work with an fMRI study. However, we would like to underscore your point regarding the importance of self-report in capturing participants’ subjective experience of emotional pain. While this study produces many questions about what is happening during consoling touch in the context of emotional pain, it is the first step towards understanding this experience. We do not believe the absence of neuroimaging data in this paper strips our findings of any value, and believe this work will be generative for future research in this area, and thus should be shared with the scientific community. 

Reviewer #3:

Comment 1: I'm fine with the current version of the manuscript. Merry Christmas, Happy Hanuka and Happy New Year!

Response 1: Thank you for your helpful feedback! Happy Holidays to you as well!

---

## [Editor Report · Decision Letter 2]

26 Jan 2021

The Comfort in Touch: Immediate and Lasting Effects of Handholding on Emotional Pain

PONE-D-20-25552R2

Dear Dr. Sahi,

We’re pleased to inform you that your manuscript has been judged scientifically suitable for publication and will be formally accepted for publication once it meets all outstanding technical requirements.

Kind regards,

Hedwig Eisenbarth

Academic Editor

PLOS ONE

Additional Editor Comments (optional):

All comments have been addressed and I accept your replies to the comments you did not agree with the reviewer.
---

## [Editor Report · Acceptance letter]

1 Feb 2021

PONE-D-20-25552R2 

The comfort in touch: Immediate and lasting effects of handholding on emotional pain 

Dear Dr. Sahi:

I'm pleased to inform you that your manuscript has been deemed suitable for publication in PLOS ONE. Congratulations! Your manuscript is now with our production department. 

Kind regards, 

on behalf of

Dr. Hedwig Eisenbarth 

Academic Editor

PLOS ONE